# CALIBRATION MATTERS: TACKLING MAXIMIZATION BIAS IN LARGE-SCALE ADVERTISING RECOMMENDATION SYSTEMS

**Yewen Fan**[*,1], **Nian Si**[*,3], **Kun Zhang**[1,2]
[1] Carnegie Mellon University
[2] Mohamed bin Zayed University of Artificial Intelligence
[3] University of Chicago Booth School of Business
([*] Equal contribution)

## ABSTRACT

Calibration is defined as the ratio of the average predicted click rate to the true click rate. The optimization of calibration is essential to many online advertising recommendation systems because it directly affects the downstream bids in ads auctions and the amount of money charged to advertisers. Despite its importance, calibration often suffers from a problem called "maximization bias". Maximization bias refers to the phenomenon that the maximum of predicted values overestimates the true maximum. The problem is introduced because the calibration is computed on the set selected by the prediction model itself. It persists even if unbiased predictions are achieved on every datapoint and worsens when covariate shifts exist between the training and test sets. To mitigate this problem, we quantify maximization bias and propose a variance-adjusting debiasing (VAD) meta-algorithm in this paper. The algorithm is efficient, robust, and practical as it is able to mitigate maximization bias problem under covariate shifts, without incurring additional online serving costs or compromising the ranking performance. We demonstrate the effectiveness of the proposed algorithm using a state-of-the-art recommendation neural network model on a large-scale real-world dataset.

## 1 INTRODUCTION

The online advertising industry has grown exponentially in the past few decades. According to Statista (2022), the total value of the global internet advertising market was worth USD 566 billion in 2020 and is expected to reach USD 700 billion by 2025.

In the online advertising industry, to help advertisers reach the target customers, demand-side platforms (DSPs) try to bid for available ad slots in an ad exchange. A DSP serves many advertisers simultaneously, and ads provided by those advertisers form the DSP's ads candidate pool. From the DSP's perspective, the advertising campaign pipeline executes as follows:

(1) The DSP uses data to build machine learning (ML) models for advertisement value estimation. An advertisement's value is often measured by the click-through rate (CTR) or conversion rate.

(2) When the ad exchange sends requests in the form of online bidding auctions for some specific ad slots to a DSP, the DSP uses the ML models to predict values for ads in its ads candidate pool.

(3) For the bidding requests, the DSP needs to choose the most suitable ads from its ads candidate pool. Therefore, based on the estimated values, the DSP chooses the ad candidates with the highest values and submits corresponding bids to the ad auctions in the ad exchange.

(4) For each auction, an ad with the highest bid would win the auction, and would be displayed (i.e., recommended) in this specific ad slot. The ad exchange would charge the winning DSP a certain amount of money based on the submitted bid and the auction mechanism.

For the machine learning models in Step (2), besides learning the ranking (i.e. which ads sent to ad exchange), DSPs also need to accurately estimate the value of the chosen ads, because in

Step (3), DSPs bid based on the estimated value obtained from Step (2). Thus, DSPs try to avoid underbidding or overbidding, the latter of which may result in over-charging advertisers. We measure the estimation accuracy by *calibration*, which is the ratio of the average estimated value (e.g., estimated click-through rate) to the average empirical value (e.g., whether user click or not).

Calibration is essential to the success of online ads bidding methods, as well-calibrated predictions are critical to the efficiency of ads auctions (He et al., 2014; McMahan et al., 2013). Calibration is also crucial in applications such as weather forecasting (Murphy & Winkler, 1977; DeGroot & Fienberg, 1983; Gneiting & Raftery, 2005), personalized medicine (Jiang et al., 2012) and natural language processing (Nguyen & O'Connor, 2015; Card & Smith, 2018). There is rich literature for model calibration methods (Zadrozny & Elkan, 2002; 2001; Menon et al., 2012; Deng et al., 2020; Naeini et al., 2015; Kumar et al., 2019; Platt et al., 1999; Guo et al., 2017; Kull et al., 2017; 2019). These existing methods focus on calibration for model bias. However, those methods do not explicitly consider the selection procedures in Step (3) of the aforementioned recommendation system pipeline. In this case, even if unbiased predictions are obtained for each ad, the calibration may perform poorly on the selection set due to *maximization bias*. Maximization bias occurs when maximization is performed on random estimated values rather than deterministic true values. We will provide a concrete example to illustrate the difference between maximization bias and model bias.

*Example* 1. Assume there are two different ads with the same "true" CTR 0.5. Now we consider an ML model that learns the CTR of the two ads from data independently. We assume that the ML model predicts the CTR of either one of the ads as 0.6 or 0.4 with equal probabilities. Note that the estimation is unbiased and thus having zero model bias. After both advertisements are submitted to the auction system, the ad with the highest estimated CTR will be selected. In this case, the probability of the system selecting an ad with an estimated CTR 0.6 is 75% and an ad with an estimated CTR 0.4 is 25%. Therefore, in this example, the model has maximization bias because it overestimates the true value of the ads ($3/4 \times 0.6 + 1/4 \times 0.4 = 0.55 > 0.5$). This example explains why there may be maximization bias in a model after selection even if the model has zero model bias.

Hypothetically, if the DSP submits all the ads with their corresponding bids to an ad exchange, the maximization bias is analogous to the so-called winner's curse, even in the absence of selection and maximization procedures during Step (3). In auction theory, the winner's curse means that in common value auctions, the winners tend to overbid if they receive noisy private signals. Consequently, this calibration issue arises in a wider context.

What makes calibration even harder is the covariate shifts between training and test data (Shen et al., 2021; Wang et al., 2021). The training data only consists of the previous winning and displayed ads, but during testing, DSPs need to select from a much larger ads candidate set. Therefore, the test set contains many ads that are underrepresented in the training set since those types of ads have never been recommended before. These covariate shifts will invalidate aforementioned calibration methods that reduce bias using labeled validation sets.

In this paper, we propose a practical meta-algorithm to tackle the maximization bias in calibration, which could be in tandem with other calibration methods. Our algorithm neither compromises the ranking performance nor increases online serving overhead (e.g., inference cost and memory cost). Our contributions are summarized below:

(1) We theoretically quantify the maximization bias in generalized linear models with Gaussian distributions. We show that the calibration error mainly depends on the variances of the predictor and the test distribution rather than number of items selected.

(2) We propose an efficient, robust, and practical meta-algorithm called variance-adjusting debias (VAD) method[1] that can apply to any machine learning method and any existing calibration methods. This algorithm is mostly executed offline without any additional online serving costs. Furthermore, the algorithm is robust to covariate shifts that are common in modern recommendation systems.

(3) We conduct extensive numerical experiments to demonstrate the effectiveness of the proposed meta-algorithm in both synthetic datasets using a logistic regression model and a large-scale real-world dataset using a state-of-the-art recommendation neural network. In particular, applying VAD in tandem with other calibration methods always improve the calibration performance compared with applying other calibration methods alone.

---

[1]code available in https://anonymous.4open.science/r/VAD

## 2 RELATED WORK

There is a long line of work regarding calibration methods. Broadly speaking, existing methods could be classified into two groups: non-parametric and parametric methods.(Kweon et al., 2021) On one hand, non-parametric methods utilizes binning ideas, which include histogram binning (Zadrozny & Elkan, 2001), isotonic regression (Zadrozny & Elkan, 2002), smoothed isotonic regression (Deng et al., 2020), Bayesian binning (Naeini et al., 2015), and scaling-binning calibrator (Kumar et al., 2019). On the other hand, parametric methods explicitly learn a parametric function mapping from the model original scores to calibrated probabilities. Example methods include Platt scaling (Platt et al., 1999), temperature scaling (Guo et al., 2017), Beta calibration (Kull et al., 2017), and Dirichlet calibration (Kull et al., 2019). We refer the readers to Kweon et al. (2021) for a comprehensive survey on various calibration methods. As discussed in Introduction, those methods are not designed for correcting maximization bias.

Maximization bias appears in many different domains, ranging from economics (Van den Steen, 2004; Capen et al., 1971), decision analysis (Smith & Winkler, 2006), to statistics, which includes model selection (Varma & Simon, 2006), over-fitting (Cawley & Talbot, 2010), selection bias (Heckman, 1979), and feature selection (Ambroise & McLachlan, 2002). Maximization bias is especially well-documented in reinforcement learning literature (see, Sutton & Barto (2018, Section 6.7)). In reinforcement learning, estimating value function is a fundamental task, where value function is typically the maximum of many expected values of different actions. To reduce maximization bias, double learning or cross-validation estimators (Van Hasselt, 2010; 2011; 2013; Van Hasselt et al., 2016) are used and demonstrate strong empirical performances. The basic idea is to train two separate models: one model selects while the other predicts the probability. However, this type of methods is not applicable to large-scale ads recommendation systems since it will double online serving costs, and online serving efficiency is essential to recommendation system performances.

Calibration for maximization bias is also closely related to estimating the maximum mean of several random variables in operations research and machine learning. Various estimators were proposed: Chen & Dudewicz (1976) develop a two-stage procedure to provide a confidence interval of the highest mean; Lesnevski et al. (2007) further integrate their method with screening ideas, variance-reduction, and common random numbers techniques; Liu et al. (2019) propose an upper confidence bound (UCB) approach; Chang et al. (2005) incorporate similar UCB components into the Monte Carlo tree search; and D'Eramo et al. (2016) use weighted average of the sample means, where the weights are computed by Gaussian approximations. However, those methods are either simulation-based or have access to multiple i.i.d. copies of random variables; thus, they cannot be directly applied to supervised learning settings in recommendation systems.

## 3 PRELIMINARIES AND PROBLEM SETTING

Consider a supervised learning setting. For each data point, we have a high-dimensional feature $X \in \mathcal{X} \subset \mathbb{R}^d$ and a label $Y \in \mathcal{Y} \triangleq \{0, 1\}$. In this paper, we focus on the binary label $Y$ that represents whether the user clicks or not. Our method can be easily extended to continuous labels. Suppose we have access to a labeled training set $(X, Y) \sim \mathcal{D}_{\text{train}}$ and an unlabeled test validation set $X \sim \mathcal{D}_{\text{val}-\text{test},X}$, which has the same distribution as the $X$ margin of the real test set $\mathcal{D}_{\text{test}}$. Note that there are often covariate shifts between $\mathcal{D}_{\text{train}}$ and $\mathcal{D}_{\text{test}}$ since the training set consists of the historical recommended items, while the test set contains all possible candidates, many of which may have never been seen by users before. However, we can reasonably assume that there are no concept drifts between training and test distributions in Assumption 3.1.

**Assumption 3.1** (No concept drift). The conditional distribution $Y$ given $X$ is the same between the training distribution and the test distribution.

Then, the recommendation system pipeline is summarized in Flowchart 1. Specifically, in the first step, the predictor $f : \mathcal{X} \rightarrow [0, 1]$ is a prediction of $\mathbb{P}(Y = 1|X)$; in the second step, we rank all items by $f(x)$ and pick the top $\alpha$ (unknown apriori) proportion, where the selected set is denoted by $\tilde{\mathcal{D}}_{\text{test}}^\alpha$; in the third step, we consider two different calibration metrics, including calibration error $\mathcal{E}$ He et al. (2014) and expected calibration error (ECE) and we provide additional results for maximum calibration error (MCE) in the Appendix (Naeini et al., 2015; Kweon et al., 2021).

The calibration error $\mathcal{E}$ and ECE are defined by

$$\mathcal{E} \triangleq \frac{\sum_{i \in \tilde{\mathcal{D}}_{\text{test}}^\alpha} f(x_i)}{\sum_{i \in \tilde{\mathcal{D}}_{\text{test}}^\alpha} y_i} - 1, \ \text{ECE} \triangleq \sum_{m=1}^M \frac{|B_m|}{N} \left| \frac{\sum_{k \in B_m} y_k}{|B_m|} - \frac{\sum_{k \in B_m} f(x_k)}{|B_m|} \right| \tag{1}$$

where we partition all items in $\tilde{\mathcal{D}}_{\text{test}}^\alpha$ into $M$ equi-spaced bins by their values, and $B_m$ is $m$-th bin and $N = |\tilde{\mathcal{D}}_{\text{test}}^\alpha|$ is the number of samples.

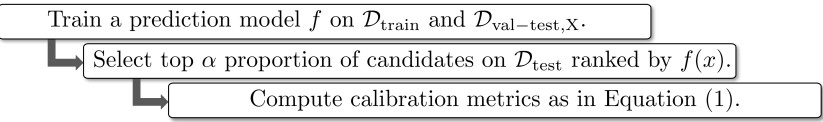

Figure 1: Flowchart visualization of the procedure.

In Section 6.2, we execute the pipeline on a real-world ads recommendation system dataset using a state-of-the-art neural network and observe that calibration errors are consistently larger than 3% as shown in Table 2, if not performing any debiasing methods. Therefore, the goal of this paper is to find a predictor $f$ that minimizes calibration errors in the selected set without compromising the ranking performance or incurring additional online serving costs. The serving costs refer to the costs of executing the method on the test sets.

We note that the unlabeled validation set $\mathcal{D}_{\text{val}-\text{test},X}$ is easily obtained and offline available because we can use the candidate sets from previous online requests. Furthermore, $\alpha$ is usually small (but unknown) in practice since DSPs usually have a large ads candidate set.

**Remark: maximization bias v.s. model bias** In this paper, we tackle maximization bias, which is different from model bias in machine learning. Model bias is the difference between the expected model prediction and true value for a given ad. Note that model predictions have randomness due to random data and stochastic optimization algorithms. However, maximization bias in our paper is a different type of bias, orthogonal to model bias. Maximization bias exists because of the selection (maximization) step. Even models with zero model bias may have maximization bias. We refer the readers back to Example 1 to see the difference between model bias and maximization bias.

## 4 Quantifying Maximization Bias in Generalized Linear Models

Generalized linear models are a unifying framework of linear regression, logistic regression, and Poisson regression (Nelder & Wedderburn, 1972). In particular, neural networks (NNs) are categorized as a generalized linear model if we view the neurons from the second-to-last layer as features and use a sigmoid function as the final activation function. In this section, we provide a rigorous quantification for the maximization bias in generalized linear models with Gaussian features.

By a slight abuse of notation, $\mathcal{D}_{\text{test}}$ represents the population distribution of $\{X, Y\}$ in the test set. We need to select the top-$\alpha$ percent from the test set. We further assume the underlying true model is generalized linear in both training and test sets with the same conditional distributions, i.e., $Y|X \sim \text{Ber}\left(\phi(\beta_*^\top X)\right)$, where $\phi(\cdot)$ is a positive, continuous differentiable, and monotonically increasing function. Let $\hat{\beta}_N$ be the parameter learned from the $N$-sample training set sampled from $\mathcal{D}_{\text{train}}$, which is independent of $\mathcal{D}_{\text{test}}$. We do not need to assume marginal distributions of $X$ in the training data $\mathcal{D}_{\text{train}}$ and test data $\mathcal{D}_{\text{test}}$ have the same distribution.

We let $q_{1-\alpha}(Z)$ be the $1 - \alpha$ quantile of distribution $Z$, i.e., $\mathbb{P}(Z \geq q_{1-\alpha}(Z)) = \alpha$.

By the monotonicity of $\phi(\cdot)$, the estimated average probability on the selection set is

$$\mathbb{E}_{\mathcal{D}_{\text{test}}, \mathcal{D}_{\text{train}}} \left[ \phi\left(\hat{\beta}_N^\top X\right) | \hat{\beta}_N^\top X \geq q_{1-\alpha}(\hat{\beta}_N^\top X | \hat{\beta}_N) \right]; \tag{2}$$

the actual average probability on the selection set is

$$\mathbb{E}_{\mathcal{D}_{\text{test}}, \mathcal{D}_{\text{train}}} \left[ \phi\left(\beta_*^\top X\right) | \hat{\beta}_N^\top X \geq q_{1-\alpha}(\hat{\beta}_N^\top X | \hat{\beta}_N) \right]. \tag{3}$$

Note that these expectations are taken with respect to the randomness of both $X$ and $\hat{\beta}_N$. To abbreviate the notion, we drop the subscripts $\mathcal{D}_{\text{test}}, \mathcal{D}_{\text{train}}$ when there is no confusion. We quantify the maximization bias for this generalized linear model with covariate shifts in Theorem 4.1.

**Theorem 4.1.** *Suppose $X \sim \mathcal{N}(\mu, \Sigma)$ and $\phi(\cdot)$ is a positive, Lipschitz continuous, twice differentiable, and monotonically increasing function. If $\hat{\beta}_N$ is a maximum likelihood estimator, we have the estimated average probability on the selection set*

$$\mathbb{E}\left[\phi\left(\hat{\beta}_N^\top X\right) | \hat{\beta}_N^\top X \geq q_{1-\alpha}(\hat{\beta}_N^\top X | \hat{\beta}_N)\right] = \mathbb{E}\left[\phi\left(\hat{\beta}_N^\top \mu + \sqrt{\hat{\beta}_N^\top \Sigma \hat{\beta}_N} Z\right) | Z \geq q_{1-\alpha}(Z)\right],$$

*and the maximization bias on the selection set is*

$$\mathbb{E}\left[\phi\left(\hat{\beta}_N^\top X\right) | \hat{\beta}_N^\top X \geq q_{1-\alpha}(\hat{\beta}_N^\top X | \hat{\beta}_N)\right] - \mathbb{E}\left[\phi\left(\beta_*^\top X\right) | \hat{\beta}_N^\top X \geq q_{1-\alpha}(\hat{\beta}_N^\top X | \hat{\beta}_N)\right]$$

$$= \mathbb{E}\left[\int_{\frac{\hat{\beta}_N^\top \Sigma \beta_*}{\sqrt{\hat{\beta}_N^\top \Sigma \hat{\beta}_N}}}^{\sqrt{\hat{\beta}_N^\top \Sigma \hat{\beta}_N}} h'(t)\mathrm{d}t\right] + O\left(\frac{1}{N}\right). \tag{4}$$

*where $Z \sim \mathcal{N}(0, 1)$ and $h'(t) \triangleq \mathbb{E}\left[\phi'\left(\beta_N^\top \mu + tZ\right) Z | Z \geq q_{1-\alpha}(Z)\right]$. The $O(1/N)$ term only depends on $\phi$, $\beta_*$, and $\Sigma$, but does not depend on $\alpha$.*

**Remark:** 1. The Gaussianity assumptions of the feature $X$ are not necessary. One only needs $\beta_*^\top X$ and $\hat{\beta}_N^\top X$ to be jointly Gaussian conditional on $\hat{\beta}_N$ and the proof would still go through. Figure 5 in Appendix B.2 shows that $\hat{\beta}_N^\top X$ is indeed very close to the Gaussian distribution.

2. Note that we do not assume that the training and test distributions are the same. Theorem 4.1 is true under arbitrary covariate shifts.

In the setting of Theorem 4.1, $\text{Var}(\hat{\beta}X | X)$ is heterogeneous across $X$, and we allow to choose any $\alpha$ proportion of the test set. Furthermore, if $\alpha$ is small and $\mu$ is small, $h'(t)$ has the same order as $\mathbb{E}[Z | Z \geq q_{1-\alpha}(Z)]$, which is large, and the maximization bias mainly depends on $\hat{\beta}_N^\top \Sigma \beta_*$ and $\hat{\beta}_N^\top \Sigma \hat{\beta}_N$. The formal statement is in Lemma A.2 in Appendix A.

To reduce the maximization bias, we discount $\hat{\beta}_N X$ by $\lambda$. Corollary 4.2 studies the bias for this discounted estimator.

**Corollary 4.2.** *Suppose the same assumptions in Theorem 4.1 are imposed. If we change $\beta_N^\top X$ to $\lambda \hat{\beta}_N^\top X + (1 - \lambda)\beta_N^\top \mu$ for $\lambda \in [0, 1]$, the bias becomes*

$$\mathbb{E}\left[\phi\left(\lambda \hat{\beta}_N^\top X + (1 - \lambda)\beta_N^\top \mu\right) | \hat{\beta}_N^\top X \geq q_{1-\alpha}(\hat{\beta}_N^\top X | \hat{\beta}_N)\right]$$

$$- \mathbb{E}\left[\phi\left(\beta_*^\top X\right) | \hat{\beta}_N^\top X \geq q_{1-\alpha}(\hat{\beta}_N^\top X | \hat{\beta}_N)\right] = \mathbb{E}\left[\int_{\frac{\hat{\beta}_N^\top \Sigma \beta_*}{\sqrt{\hat{\beta}_N^\top \Sigma \hat{\beta}_N}}}^{\lambda\sqrt{\hat{\beta}_N^\top \Sigma \hat{\beta}_N}} h'(t)\mathrm{d}t\right] + O\left(\frac{1}{N}\right). \tag{5}$$

The transformation $\hat{\beta}_N^\top X \mapsto \lambda \hat{\beta}_N^\top X + (1 - \lambda)\beta_N^\top \mu$ is linear; thus, it does not alter the item rankings. Despite its simplicity, we are able to find a $\lambda$ such that the leading term of (5) is approximately zero in Section 5, i.e.,

$$\mathbb{E}\left[\int_{\frac{\hat{\beta}_N^\top \Sigma \beta_*}{\sqrt{\hat{\beta}_N^\top \Sigma \hat{\beta}_N}}}^{\lambda\sqrt{\hat{\beta}_N^\top \Sigma \hat{\beta}_N}} h'(t)\mathrm{d}t\right] \approx 0.$$

## 5 VARIANCE-ADJUSTING DEBIASING META-ALGORITHM

Based on the theory developed in Section 4, the goal is to find $\lambda$ such that

$$\lambda\sqrt{\hat{\beta}_N^\top \Sigma \hat{\beta}_N} \approx \frac{\hat{\beta}_N^\top \Sigma \beta_*}{\sqrt{\hat{\beta}_N^\top \Sigma \hat{\beta}_N}} \Leftrightarrow \lambda \approx \frac{\hat{\beta}_N^\top \Sigma \beta_*}{\hat{\beta}_N^\top \Sigma \hat{\beta}_N}.$$

The denominator $\hat{\beta}_N^\top \Sigma \hat{\beta}_N$ is easy to estimate as $\hat{\beta}_N^\top \Sigma \hat{\beta}_N = \text{Var}_{\mathcal{D}_{\text{test}}}(\hat{\beta}_N^\top X | \hat{\beta}_N)$. To estimate the nominator $\hat{\beta}_N^\top \Sigma \beta_*$, we first observe the decomposition that

$$\mathbb{E}[\hat{\beta}_N^\top \Sigma \hat{\beta}_N] = \mathbb{E}[\hat{\beta}_N^\top \Sigma \beta_*] + \mathbb{E}_{\mathcal{D}_{\text{test}}}\left[\text{Var}_{\mathcal{D}_{\text{train}}}\left(\hat{\beta}_N^\top X - \hat{\beta}_N^\top \mu | X\right)\right] \tag{6}$$

provided that $\mathbb{E}[\hat{\beta}_N] = \beta_*$; this equality is proved in Appendix A. Therefore, $\hat{\beta}_N^\top \Sigma \beta_*$ could be approximated by

$$\hat{\beta}_N^\top \Sigma \hat{\beta}_N - \mathbb{E}_{\mathcal{D}_{\text{test}}} \left[ \text{Var}_{\mathcal{D}_{\text{train}}} \left( \hat{\beta}_N^\top X - \hat{\beta}_N^\top \mu | X \right) \right],$$

where the variance of $\hat{\beta}_N$ could be estimated by bootstrapping or retraining the model using different random seeds. Note that this approximation is relatively accurate if the feature dimension $d$ is large and the correlations between dimensions are small. This is because in this case, $\hat{\beta}_N^\top \Sigma \hat{\beta}_N$ and $\hat{\beta}_N^\top \Sigma \beta_*$ concentrate on their means with an error $O_p(1/\sqrt{d})$:

$$\hat{\beta}_N^\top \Sigma \hat{\beta}_N = \mathbb{E}[\hat{\beta}_N^\top \Sigma \hat{\beta}_N] + O_p\left(\frac{1}{\sqrt{d}}\right) \text{ and } \hat{\beta}_N^\top \Sigma \beta_* = \mathbb{E}[\hat{\beta}_N^\top \Sigma \beta_*] + O_p\left(\frac{1}{\sqrt{d}}\right).$$

In the context of generalized linear models, we observe that $\hat{\beta}_N^\top X = \phi^{-1}(f)$. Therefore, in general cases, we define $l(x) = \phi^{-1}(f(x))$. In practice, $l(x)$ can be obtained either by inverting $\phi$ or extracting the last layer of the NN if the neural network is used as the prediction model with $\phi$ as the activation function of the last layer. Then, based on these analysis, we propose a variance-adjusting debiasing (VAD) meta-algorithm in Algorithm 1. Since the $\lambda$ estimation procedure is fully non-parametric and depends only on some means, variances, and conditional variances, our meta-algorithm applies to any machine learning algorithm.

---

**Algorithm 1** Variance-adjusting debiasing (VAD) method

---

1: **Input:** Training dataset $\mathcal{D}_{\text{train}}$, the unlabeled test validation set $\mathcal{D}_{\text{val}-\text{test},X}$, a link function $\phi : \mathbb{R} \to [0, 1]$, and the number of replications $S$.
2: **Output:** A variance adjusting debiased predictor $f_{\text{VAD}}$.
3: Train a model on the training set and obtain the predictor $f_1$.
4: Bootstrap (i.e. random sampling with replacement) the dataset $S - 1$ times or retrain the model $S - 1$ times using different random seeds and obtain the predictor $f_2, \ldots, f_S$.
5: Let $l_i(x) = \phi^{-1}(f_i(x))$ for $i \in [S]$. Then compute the means $\bar{Y}_i^l = \mathbb{E}_{\mathcal{D}_{\text{val}-\text{test},X}}[l_i(X)]$ for $i \in [S]$ the test variance $\left(\hat{\sigma}_{\hat{Y}}^l\right)^2 = \text{Var}_{\mathcal{D}_{\text{val}-\text{test},X}}[l_1(X)]$.
6: Compute the expected conditional variance

$$\left(\hat{\sigma}_f^l\right)^2 = \mathbb{E}_{\mathcal{D}_{\text{val}-\text{test},X}} \left[ \frac{1}{S-1} \sum_{j=1}^{S} \left( l_j(X) - \bar{Y}_j^l - \frac{1}{S} \left( \sum_{i=1}^{S} \left( l_i(X) - \bar{Y}_i^l \right) \right) \right)^2 \right].$$

7: Compute $\lambda = 1 - \left(\hat{\sigma}_f^l\right)^2 / \left(\hat{\sigma}_{\hat{Y}}^l\right)^2$.
8: Output $f_{\text{VAD}}(\cdot) = \phi\left(\lambda l_1(\cdot) + (1 - \lambda)\bar{Y}_1^l\right)$.

---

Since the debiased predictor $f_{\text{VAD}}$ is a monotonic transformation of the original predictor $f_1$ in Algorithm 1, the prediction rankings remain the same. Further, $\lambda$ and $\bar{Y}_1^l$ are computed purely offline as long as we have a unlabeled validation set that has the same distribution as the test set. Thus, no additional serving costs are added. Furthermore, since we only need to estimate means and variances, we only need samples from the unlabeled candidate set of reasonable size. We can sample recent data points in the large (and potentially non-stationary) candidate set to get sufficiently accurate estimates.

For the link function $\phi(\cdot)$ in logistic regression and NNs with a final sigmoid activation function, $\phi(x) = (1 + \exp(-x))^{-1}$ and $l_i$ is the logit (the last layer in NNs) of the predictor $f_i$.

$\phi(\cdot)$ could also be chosen as the identity mapping $\phi(x) = x$. We note that this choice of $\phi(\cdot)$ has similar performance to the choice of $\phi(x) = (1 + \exp(-x))^{-1}$. We report additional results about the behavior of the identity link function in Appendix B.

On Line 4 in Algorithm 1, we recommend bootstrapping (Efron & Tibshirani, 1994) if the base training model lacks intrinsic randomness, e.g., logistic regression, which is an efficiently solvable convex optimization problem. However, if the base model is highly non-convex with multiple local optima, e.g., neural networks, we recommend random initializations and random data orders because many empirical studies (Nixon et al., 2020; Lakshminarayanan et al., 2016; Lee et al., 2015) show that

bootstrapping may hurt the performance in deep neural networks and the estimation of the conditional variances would benefit from the algorithmic randomness (Jiang et al., 2021).

In our method, we only need to choose one hyper-parameter $S$ and we do not need to specify $\alpha$. In fact, $S = 2$ would be sufficient and results in lower training cost. Therefore, our method doubles the training cost, which is usually acceptable in practice. More importantly, our method does not incur any additional online serving costs. All results reported in Section 6 use $S = 2$. We report additional results about different $S$ choices in Appendix B.

## 6 EXPERIMENT RESULTS

In this section, we demonstrate the performance of our method in both synthetic data and a real-world ads recommendation dataset. We use calibration errors, ECE, and MCE to evaluate performances. Note that evaluation metrics are calculated on the selection set, i.e., we choose top $\alpha$ proportion of test data points using model predictions and compute the evaluation metrics in the top-$\alpha$ selection set. We provide additional numerical results for the Avazu dataset [2] in Appendix B.3.

### 6.1 SYNTHETIC DATA

**Data and Model** We consider a logistic regression model. We assume the response $Y$ follows the Bernoulli distribution with probability $\left(1 + \exp(-\beta^\top X)\right)^{-1}$, for $\beta, X \in \mathbb{R}^d$.

In the experiments, we consider $d = 20$, $\beta = [1, 1, \ldots, 1]^\top$, $X \sim \mathcal{N}(\mu_{\text{train}}, \Sigma_{\text{train}})$ in the training set $\mathcal{D}_{\text{train}}$, and $X \sim \mathcal{N}(\mu_{\text{test}}, \Sigma_{\text{test}})$ in the test set $\mathcal{D}_{\text{test}}$. In the training set, we generate $N_{\text{train}} = 3000$ i.i.d. training samples to train a logistic regression model. For our proposed method, we bootstrap $S = 2$ times. In the test set, we choose top $\alpha$ proportion of $N_{\text{test}} = 30000$ data points using the model predictions and compute the evaluation metrics in the top-$\alpha$ selection set. We generate $N_{\text{val}-\text{test}} = 30000$ data points as the test validation dataset.

**Covariate Shift** We assume there is a covariate shift and no concept drifts, where we assume

$$\mu_{\text{train}} = [0.05, 0.05, \ldots, 0.05]^\top, \text{ and } \mu_{\text{test}} = [-0.05, -0.05, \ldots, -0.05]^\top,$$

and $\Sigma_{\text{train}} = \Sigma_{\text{test}} = 0.1^2 \times I_{d \times d}$. The positive sample rate is 72.3% in the training set, which is higher than the positive sample ratio in the test set (27.7%). This is consistent with real-world recommendation systems since the training data is the previously recommended sets, in which items should have better performance than items in the whole candidate sets (i.e. test sets).

**Performance** We replicate the experiments 100 times and report averages and standard errors of calibration errors and ECE (with number of bins $M = 10$) for $\alpha \in \{2\%, 10\%\}$ in Table 1. (MCE is in Appendix B.1), where `Vanilla` stands for original predictions. We plot average calibration errors and average ECE for $\alpha \in [2\%, 10\%]$ in Appendix B.1. We also report Log Loss improvement in Appendix B.1, indicating our method improves prediction quality as well.

Table 1: Average and standard errors of calibration errors and ECE for synthetic data

| Method | $\alpha = 2\%$ | | $\alpha = 10\%$ | |
| | Calibration error | ECE | Calibration error | ECE |
| --- | --- | --- | --- | --- |
| Vanilla | 8.55%±0.68% | 0.0656±0.0019 | 7.34%±0.75% | 0.0425±0.0021 |
| VAD | **0.06%**±0.72% | **0.0572**±0.0013 | **0.62%**±0.73% | **0.0334**±0.0015 |

Note that we do not compare with other calibration methods under a logistic regression model since logistic regression produces well-calibrated predictions. Applying other calibration methods to a logistic regression model will not improve the performance. (Niculescu-Mizil & Caruana, 2005).

Table 1 shows that the vanilla model without debiasing has a calibration error of more than 7%, which is mainly due to maximization bias as logistic regression produces well-calibrated predictions. After applying VAD, the calibration error is sufficiently close to zero. All the superiority is statistically significant at the 1% significance level. Additional experiment results are reported in Appendix B.1.

---

[2] https://www.kaggle.com/c/avazu-ctr-prediction

## 6.2 REAL-WORLD DATA

**Dataset** We use the Criteo Ad Kaggle dataset [3] to demonstrate our method's performance. The Criteo Ad Kaggle dataset is a common benchmark dataset for CTR predictions. It consists of a week's worth of data, approximately 45 million samples in total. Each data point contains a binary label, which indicates whether the user clicks or not, along with 13 continuous, 26 categorical features. The positive label accounts for 25.3% of all data. The categorical features consist of 1.3 million categories on average, with 1 feature having more than 10 million categories, 5 features having more than 1 million categories. Due to computational constraints in our experiments, we use the first 15 million samples, shuffle the dataset randomly, and split the whole dataset into 85% train $\mathcal{D}_{\text{train}}$, 1.5% validation-train $\mathcal{D}_{\text{val-train}}$, 1.5% validation-test $\mathcal{D}_{\text{val-test}}$, and 12% test $\mathcal{D}_{\text{test}}$ datasets.

**Base Model** We use the state-of-the-art deep learning recommendation model (DLRM) (Naumov et al., 2019) open-sourced by Meta as our baseline model. DLRM employs a standard architecture for ranking tasks, with embeddings to handle categorical features, Multilayer perceptrons (MLPs) to handle continuous features and the interactions of categorical features and continuous features. Throughout our experiments, we use the default parameters and a SGD optimizer. Note that our method is model-agnostic, so it can be directly applied to other models (e.g. support vector machines, boosted trees, nearest neighbors, etc Friedman et al. (2001)).

**Baseline Calibration Methods** We compare our method with various classic calibration methods. For parametric methods, we compare against Platt scaling (Platt et al., 1999). For non-parametric methods, we compare against histogram binning (Zadrozny & Elkan, 2002; 2001), isotonic regression (Menon et al., 2012), and scaling-binning calibrator (Kumar et al., 2019). We use the labeled training validation dataset to do calibration for above methods.

Note that none of the existing work explicitly considers maximization bias and thus fails to perform well in our setting. VAD can be combined with all existing calibration methods to achieve better performance by making a small change to the original VAD algorithm. VAD takes predictions calibrated by other calibration methods as inputs. Additionally, instead of directly using $\lambda$ calculated from $\mathcal{D}_{\text{val-test},X}$, we first calculate $\lambda_{\text{val-test}}$ from $\mathcal{D}_{\text{val-test},X}$ using original predictions and $\lambda_{\text{val-train}}$ from $\mathcal{D}_{\text{val-train},X}$ (the unlabeled validation set which has the same distribution as the X margin of the training set) using original predictions, and then use $\lambda = \lambda_{\text{val-test}}/\lambda_{\text{val-train}}$ to adjust predictions calibrated by other methods. This change is due to the fact that other calibration methods already compensate for maximization bias in training distribution to some extent.

**VAD Parameters** The last layer of the DLRM network uses the sigmoid activation function. In our method, by using the link function $\phi(x) = (1 + \exp(-x))^{-1}$, we compute the means $\bar{Y}_i^l$, variance $(\hat{\sigma}_{\hat{Y}}^l)^2$, and expected conditional variance $(\hat{\sigma}_f^l)^2$ of the last layer's neuron. To compute the expected prediction variance $(\hat{\sigma}_f^l)^2$, we keep the training data unchanged and modify the random initialization and data orders since the optimizer itself incurs sufficient randomness. In the experiment, we train $S = 2$ times for our method.

**Covariate Shift** Since the underlying true data-generating process is unknown, we employ a different strategy to construct out-of-distribution test data than how we generate synthetic data: we train another DLRM model using $85\% \times 15$ million samples different from the original dataset; we randomly keep each data point in the original test set with probability $1 - p$, where $p$ is the newly-trained DLRM model prediction for the data point. The training set remains the same. After this shift, the positive samples account for 20.1% of all test data. By doing this, we ensure that the distributional change is only a covariate shift, and the positive sample ratio in the test data is lower than the positive sample ratio in the training data, which is consistent with the real-world recommendation systems.

**Performance** We replicate the experiments 40 times and report averages and standard errors of calibration errors and ECE (with number of bins $M = 50$) for $\alpha \in \{2\%, 10\%\}$ in Tables 2 and 3. (MCE is in Appendix B.2), where `Original` stands for using calibration methods solely and `VAD+` represents the tandem combination of the calibration methods and VAD. We plot average calibration errors and average ECE for $\alpha \in [2\%, 10\%]$ in Figure 2. We also report Log Loss improvement in Appendix B.2, indicating our method also improves prediction quality.

---

[3]https://www.kaggle.com/c/criteo-display-ad-challenge

Table 2: Average calibration errors on the Criteo Ad Kaggle dataset

|  | $\alpha = 2\%$ | | $\alpha = 10\%$ | |
| Method | Original | VAD+ | Original | VAD+ |
| --- | --- | --- | --- | --- |
| Vanilla | 3.23%±0.41% | **-0.44%**±0.46% | 3.94%±0.63% | **-0.47%**±0.66% |
| Histogram Binning | 2.18%±0.06% | 1.44%±0.06% | 1.87%±0.04% | 0.98%±0.04% |
| Platt Scaling | 1.60%±0.13% | 0.86%±0.12% | 1.80%±0.07% | 0.91%±0.07% |
| Scaling-Binning | 2.30%±0.13% | 1.56%±0.12% | 1.99%±0.07% | 1.11%±0.07% |
| Isotonic Regression | 1.62%±0.06% | 0.87%±0.07% | 1.75%±0.04% | 0.85%±0.04% |

Table 3: Average ECE on the Criteo Ad Kaggle dataset

|  | $\alpha = 2\%$ | | $\alpha = 10\%$ | |
| Method | Original | VAD+ | Original | VAD+ |
| --- | --- | --- | --- | --- |
| Vanilla | 0.0287±0.0024 | 0.0225±0.0016 | 0.0271±0.0029 | 0.0208±0.0022 |
| Histogram Binning | 0.0213±0.0003 | 0.0185±0.0003 | 0.0132±0.0002 | 0.0105±0.0002 |
| Platt Scaling | 0.0179±0.0006 | **0.0158**±0.0004 | 0.0121±0.0003 | **0.0092**±0.0002 |
| Scaling-Binning | 0.0217±0.0007 | 0.0188±0.0005 | 0.0131±0.0003 | 0.0101±0.0002 |
| Isotonic Regression | 0.0193±0.0004 | 0.0175±0.0003 | 0.0126±0.0002 | 0.0101±0.0002 |

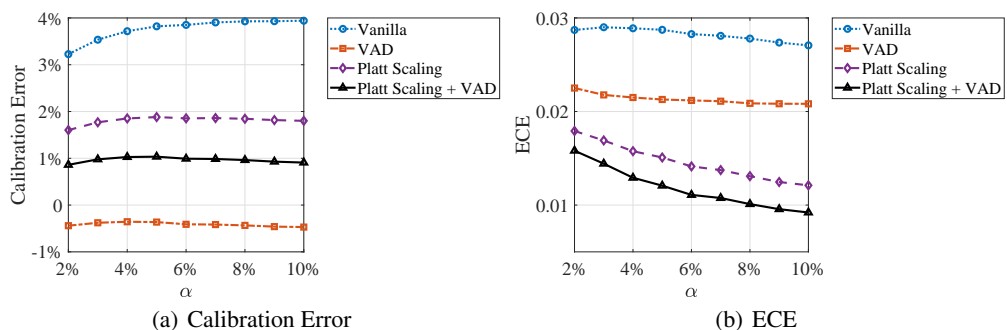

(a) Calibration Error          (b) ECE

Figure 2: Average calibration errors and ECE for the Criteo Ad Kaggle dataset

Table 2 and Figure 2(a) show that the vanilla model without debiasing has a calibration error more than 3%. After debiased by existing calibration methods, there are still $1.5\% \sim 2.0\%$ over-calibrations, which are largely due to maximization bias. Among all methods, the sole VAD method (i.e., not functioning in tandem with any other calibration methods) perform the best, but has a large variance. Moreover, from Tables 2, 3 and Figure 2, We find that our method (VAD) outperforms the vanilla method. Particularly, all the calibration methods in tandem with VAD achieve better performance than the sole calibration methods alone. All the superiority is statistically significant at the 1% significance level. Additional experiment results are reported in Appendix B.2.

## 7 CONCLUSION

We proposed a theory-certified meta-algorithm variance-adjusting debiasing (VAD) to tackle maximization bias in recommendation systems. The meta-algorithm is easy-to-implement by adding only a few lines, scalable to large-scale systems with no additional serving costs, applicable to any machine learning methods, and robust to covariate shifts between training and test sets. Empirical results show its significant superiority over other methods. Our method can be directly used in industry with minor modifications, e.g. doing VAD separately for each group of data instead of doing VAD globally. Interesting follow-ups include combining VAD and other calibration methods in a better way and further reducing the training cost. We leave these for future work.

## 8 REPRODUCIBILITY STATEMENT

We open-sourced our implementation at `https://github.com/tofuwen/VAD`.

## 9 ACKNOWLEDGEMENTS

This project was partially supported by the National Institutes of Health (NIH) under Contract R01HL159805, by the NSF-Convergence Accelerator Track-D award #2134901, by a grant from Apple Inc., a grant from KDDI Research Inc, and generous gifts from Salesforce Inc., Microsoft Research, and Amazon Research.

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

## A  PROOFS

Before the proof of Theorem 4.1, we first collect some useful results from the standard MLE theory.

**Lemma A.1.** $\hat{\beta}$ *follows the central limit theorem:*

$$\sqrt{N}\left(\hat{\beta}_N - \beta_*\right) \Rightarrow N(0, \mathcal{I}^{-1}),$$

*where $\mathcal{I}$ is the Fisher information matrix defined as*

$$\mathcal{I}_{jk} = \mathbb{E}_{\mathcal{D}_{\text{train}}}\left[-\frac{\partial^2\left(Y\ln\left(\phi\left(\beta_*^\top X\right)\right) + (1-Y)\ln\left(1 - \phi\left(\beta_*^\top X\right)\right)\right)}{\partial\beta_j\partial\beta_k}\right].$$

*Furthermore, the bias of $\hat{\beta}$ is of order $1/N$, i.e., $\mathbb{E}[\hat{\beta}_N - \beta_*] = O(1/N)$.*

*Proof.* The first claim follows from the Lipschitzness of the link function $\phi$ and Theorem 5.39 in Van der Vaart (2000). The second claim follows from formula 20 in Cox & Snell (1968). $\qquad\square$

Let $Z \sim \mathcal{N}(0,1)$, then conditional on $\hat{\beta}_N^\top$, we have $\hat{\beta}_N^\top X \sim \mathcal{N}\left(\hat{\beta}_N^\top\mu, \hat{\beta}_N^\top\Sigma\hat{\beta}_N\right)$ and $\beta_*^\top X \sim \mathcal{N}\left(\beta_*^\top\mu, \beta_*^\top\Sigma\beta_*\right)$ the standard normal distribution. Then, we have for the estimated average probability on the selection set conditional on $\hat{\beta}_N^\top$,

$$\mathbb{E}_{\mathcal{D}_{\text{test}}}\left[\phi\left(\hat{\beta}_N^\top X\right) | \hat{\beta}_N^\top X \geq q_{1-\alpha}(\hat{\beta}_N^\top X | \hat{\beta}_N), \hat{\beta}_N\right] = \mathbb{E}\left[\phi\left(\hat{\beta}_N^\top\mu + \sqrt{\hat{\beta}_N^\top\Sigma\hat{\beta}_N}Z\right) | Z \geq q_{1-\alpha}(Z), \hat{\beta}_N\right],$$

Note that conditional on $\hat{\beta}_N^\top$, $\text{Cov}_{\mathcal{D}_{\text{test}}}\left(\hat{\beta}_N^\top X, \beta_*^\top X | \hat{\beta}_N\right) = \hat{\beta}_N^\top\Sigma\beta_*$. Therefore, we have

$$\beta_*^\top X = \frac{\hat{\beta}_N^\top\Sigma\beta_*}{\hat{\beta}_N^\top\Sigma\hat{\beta}_N}\hat{\beta}_N^\top X + \left(\beta_*^\top X - \frac{\hat{\beta}_N^\top\Sigma\beta_*}{\hat{\beta}_N^\top\Sigma\hat{\beta}_N}\hat{\beta}_N^\top X\right),$$

where

$$\left(\beta_*^\top X - \frac{\hat{\beta}_N^\top\Sigma\beta_*}{\hat{\beta}_N^\top\Sigma\hat{\beta}_N}\hat{\beta}_N^\top X\right) \perp \hat{\beta}_N^\top X \bigg| \hat{\beta}_N, \beta_*.$$

Note that

$$\beta_*^\top X - \frac{\hat{\beta}_N^\top\Sigma\beta_*}{\hat{\beta}_N^\top\Sigma\hat{\beta}_N}\hat{\beta}_N^\top X \bigg| \hat{\beta}_N, \beta_* \sim \mathcal{N}\left(\beta_*^\top\mu - \frac{\hat{\beta}_N^\top\Sigma\beta_*}{\hat{\beta}_N^\top\Sigma\hat{\beta}_N}\hat{\beta}_N^\top\mu, \beta_*^\top\Sigma\beta_* - \frac{\left(\hat{\beta}_N^\top\Sigma\beta_*\right)^2}{\hat{\beta}_N^\top\Sigma\hat{\beta}_N}\right)$$

then, we have

$$\beta_*^\top X \stackrel{d}{=} \frac{\hat{\beta}_N^\top\Sigma\beta_*}{\sqrt{\hat{\beta}_N^\top\Sigma\hat{\beta}_N}}Z + \sqrt{\beta_*^\top\Sigma\beta_* - \frac{\left(\hat{\beta}_N^\top\Sigma\beta_*\right)^2}{\hat{\beta}_N^\top\Sigma\hat{\beta}_N}}Z_2 + \beta_*^\top\mu,$$

where $Z_2 \sim \mathcal{N}(0,1)$ independent to $Z$ and $\stackrel{d}{=}$ means equal in distribution. Thus, the actual average probability can be reformulated as

$$\mathbb{E}_{\mathcal{D}_{\text{test}}}\left[\phi\left(\beta_*^\top X\right) | \hat{\beta}_N^\top X \geq q_{1-\alpha}(\hat{\beta}_N^\top X | \hat{\beta}_N), \hat{\beta}_N\right]$$

$$=\mathbb{E}\left[\phi\left(\frac{\hat{\beta}_N^\top\Sigma\beta_*}{\sqrt{\hat{\beta}_N^\top\Sigma\hat{\beta}_N}}Z + \sqrt{\beta_*^\top\Sigma\beta_* - \frac{\left(\hat{\beta}_N^\top\Sigma\beta_*\right)^2}{\hat{\beta}_N^\top\Sigma\hat{\beta}_N}}Z_2 + \beta_*^\top\mu\right) \bigg| Z \geq q_{1-\alpha}(Z), \hat{\beta}_N\right].$$

Note that $\hat{\beta}_N$ only depends on the training set. Therefore, $\hat{\beta}_N, Z, Z_2$ are mutually independent. By the Taylor expansion, we have

$$
\mathbb{E}_{\mathcal{D}_{\text{train}}}\left[\phi\left(\frac{\hat{\beta}_N^\top \Sigma \beta_*}{\sqrt{\hat{\beta}_N^\top \Sigma \hat{\beta}_N}} Z + \sqrt{\beta_*^\top \Sigma \beta_* - \frac{\left(\hat{\beta}_N^\top \Sigma \beta_*\right)^2}{\hat{\beta}_N^\top \Sigma \hat{\beta}_N}} Z_2 + \beta_*^\top \mu\right)\Bigg| Z\right]
$$

$$
= \mathbb{E}_{\mathcal{D}_{\text{train}}}\left[\phi\left(\frac{\hat{\beta}_N^\top \Sigma \beta_*}{\sqrt{\hat{\beta}_N^\top \Sigma \hat{\beta}_N}} Z + \beta_*^\top \mu\right)\Bigg| Z\right]
$$

$$
+ \mathbb{E}_{\mathcal{D}_{\text{train}}}\left[\phi'\left(\frac{\hat{\beta}_N^\top \Sigma \beta_*}{\sqrt{\hat{\beta}_N^\top \Sigma \hat{\beta}_N}} Z + \beta_*^\top \mu\right)\sqrt{\beta_*^\top \Sigma \beta_* - \frac{\left(\hat{\beta}_N^\top \Sigma \beta_*\right)^2}{\hat{\beta}_N^\top \Sigma \hat{\beta}_N}}\Bigg| Z\right]\mathbb{E}\left[Z_2\right] + O\left(\frac{1}{N}\right)
$$

$$
= \mathbb{E}_{\mathcal{D}_{\text{train}}}\left[\phi\left(\frac{\hat{\beta}_N^\top \Sigma \beta_*}{\sqrt{\hat{\beta}_N^\top \Sigma \hat{\beta}_N}} Z + \beta_*^\top \mu\right)\Bigg| Z\right] + O\left(\frac{1}{N}\right).
$$

By Lemma A.1, we have

$$
\mathbb{E}_{\mathcal{D}_{\text{train}}}\left[\left(\beta_*^\top \Sigma \beta_* - \frac{\left(\hat{\beta}_N^\top \Sigma \beta_*\right)^2}{\hat{\beta}_N^\top \Sigma \hat{\beta}_N}\right)\right] = O\left(\frac{1}{N}\right).
$$

Finally by taking the Taylor expansion of $\phi\left(\hat{\beta}_N^\top \mu + \sqrt{\hat{\beta}_N^\top \Sigma \hat{\beta}_N} Z\right)$ around $\frac{\hat{\beta}_N^\top \Sigma \beta_*}{\sqrt{\hat{\beta}_N^\top \Sigma \hat{\beta}_N}} Z + \beta_*^\top \mu$, we have

$$
\mathbb{E}_{\mathcal{D}_{\text{train}}}\left[\phi\left(\frac{\hat{\beta}_N^\top \Sigma \beta_*}{\sqrt{\hat{\beta}_N^\top \Sigma \hat{\beta}_N}} Z + \beta_*^\top \mu\right)\Bigg| Z\right] = \mathbb{E}_{\mathcal{D}_{\text{train}}}\left[\phi\left(\frac{\hat{\beta}_N^\top \Sigma \beta_*}{\sqrt{\hat{\beta}_N^\top \Sigma \hat{\beta}_N}} Z + \beta_N^\top \mu\right)\Bigg| Z\right] + O\left(\frac{1}{N}\right).
$$

Therefore, the actual average probability is

$$
\mathbb{E}\left[\phi\left(\beta_*^\top X\right)|\hat{\beta}_N^\top X \geq q_{1-\alpha}(\hat{\beta}_N^\top X|\hat{\beta}_N)\right] = \mathbb{E}\left[\phi\left(\frac{\hat{\beta}_N^\top \Sigma \beta_*}{\sqrt{\hat{\beta}_N^\top \Sigma \hat{\beta}_N}} Z + \beta_N^\top \mu\right)\Bigg| Z \geq q_{1-\alpha}(Z)\right].
$$

Let

$$
h(t) = \mathbb{E}\left[\phi\left(\beta_N^\top \mu + tZ\right)|Z \geq q_{1-\alpha}(Z)\right],
$$

and

$$
h'(t) = \mathbb{E}\left[\phi'\left(\beta_N^\top \mu + tZ\right)Z|Z \geq q_{1-\alpha}(Z)\right].
$$

Then, by using the Taylor expansion again, the maximization bias is

$$
\mathbb{E}\left[\phi\left(\hat{\beta}_N^\top X\right)|\hat{\beta}_N^\top X \geq q_{1-\alpha}(\hat{\beta}_N^\top X|\hat{\beta}_N)\right] - \mathbb{E}\left[\phi\left(\beta_*^\top X\right)|\hat{\beta}_N^\top X \geq q_{1-\alpha}(\hat{\beta}_N^\top X|\hat{\beta}_N)\right]
$$

$$
= \mathbb{E}\left[\phi\left(\hat{\beta}_N^\top \mu + \sqrt{\hat{\beta}_N^\top \Sigma \hat{\beta}_N} Z\right)|Z \geq q_{1-\alpha}(Z)\right]
$$

$$
- \mathbb{E}\left[\phi\left(\frac{\hat{\beta}_N^\top \Sigma \beta_*}{\sqrt{\hat{\beta}_N^\top \Sigma \hat{\beta}_N}} Z + \beta_N^\top \mu\right)\Bigg| Z \geq q_{1-\alpha}(Z)\right] + O\left(\frac{1}{N}\right)
$$

$$
= \mathbb{E}\left[\int_{\frac{\hat{\beta}_N^\top \Sigma \beta_*}{\sqrt{\hat{\beta}_N^\top \Sigma \hat{\beta}_N}}}^{\sqrt{\hat{\beta}_N^\top \Sigma \hat{\beta}_N}} h'(t)\mathrm{d}t\right] + O\left(\frac{1}{N}\right).
$$

*Proof of Corollary 4.2.* Conditional on $\hat{\beta}_N$, we have $\lambda\hat{\beta}_N^\top X + (1-\lambda)\hat{\beta}_N^\top\mu \sim \mathcal{N}\left(\hat{\beta}_N^\top\mu, \lambda^2\hat{\beta}_N^\top\Sigma\hat{\beta}_N\right)$. Then, we have

$$\mathbb{E}_{\mathcal{D}_{\text{test}}}\left[\phi\left(\lambda\hat{\beta}_N^\top X + (1-\lambda)\hat{\beta}_N^\top\mu\right)|\hat{\beta}_N^\top X \geq q_{1-\alpha}(\hat{\beta}_N^\top X|\hat{\beta}_N), \hat{\beta}_N\right]$$
$$= \mathbb{E}\left[\phi\left(\hat{\beta}_N^\top\mu + \lambda\sqrt{\hat{\beta}_N^\top\Sigma\hat{\beta}_N}Z\right)|Z \geq q_{1-\alpha}(Z), \hat{\beta}_N\right].$$

The remaining proof follows similar lines with the proof of Theorem 4.1. □

*Proof of Equation 6.* Note that $\mathbb{E}\left[\hat{\beta}_N\right] = \beta_*$, we have

$$\mathbb{E}[\hat{\beta}_N^\top\Sigma\hat{\beta}_N] = \mathbb{E}[\hat{\beta}_N^\top\Sigma\beta_*] + \mathbb{E}[\hat{\beta}_N^\top\Sigma(\hat{\beta}_N - \beta_*)]$$
$$= \mathbb{E}[\hat{\beta}_N^\top\Sigma\beta_*] + \mathbb{E}[\left(\hat{\beta}_N^\top - \beta_*\right)\Sigma(\hat{\beta}_N - \beta_*)].$$

Then, the second term on the right hand size is equivalent to

$$\mathbb{E}[\left(\hat{\beta}_N^\top - \beta_*\right)\Sigma(\hat{\beta}_N - \beta_*)] = \mathbb{E}[\left(\hat{\beta}_N^\top - \beta_*\right)^\top(X-\mu)(X-\mu)^\top(\hat{\beta}_N - \beta_*)]$$
$$= \mathbb{E}\left[\left([(X-\mu)^\top(\hat{\beta}_N - \beta_*)\right)^2\right].$$

By taking expectation conditional on $X$, we have

$$\mathbb{E}_{\mathcal{D}_{\text{train}}}\left[\left([(X-\mu)^\top(\hat{\beta}_N - \beta_*)\right)^2|X\right] = \text{Var}_{\mathcal{D}_{\text{train}}}\left((X-\mu)^\top\hat{\beta}_N|X\right),$$

which is because $\mathbb{E}\left[\hat{\beta}_N\right] = \beta_*$. By the tower property, we have the desired result. □

**Lemma A.2.** *We assume*

1. *$\phi'(x) \leq C$ for $x \in \mathbb{R}$ and $\phi'(x) \geq c_0 > 0$ for $x \in [l, r]$.*

2. *$\mathbb{P}\left(\beta_N^\top\mu \in [\mu_l, \mu_r]\right) \geq c_1 > 0$ and $l \leq \mu_l + t_l q_{1-\alpha}(Z) \leq \mu_r + 2t_r q_{1-\alpha}(Z) \leq r$ for some $t_l < t_r$.*

*Then, we have for $t \in [t_l, t_r]$ and $\alpha \leq 0.2$,*
$$\frac{c_0 c_1}{2}\mathbb{E}\left[Z|Z \geq q_{1-\alpha}(Z)\right] \leq h'(t) \leq C\mathbb{E}\left[Z|Z \geq q_{1-\alpha}(Z)\right].$$

*Proof.* The upper bound is immediate as $\phi'\left(\beta_N^\top\mu + tZ\right) \leq C$ and

$$h'(t) = \mathbb{E}\left[\phi'\left(\beta_N^\top\mu + tZ\right)Z|Z \geq q_{1-\alpha}(Z)\right]$$
$$\leq C\mathbb{E}\left[Z|Z \geq q_{1-\alpha}(Z)\right].$$

Now, we focus on the lower bound.
$$h'(t) \geq \mathbb{E}\left[\phi'\left(\beta_N^\top\mu + tZ\right)Z\mathbb{I}\{\beta_N^\top\mu \in [\mu_l, \mu_r]\}|Z \geq q_{1-\alpha}(Z)\right]$$
$$\geq \mathbb{E}\left[\phi'\left(\beta_N^\top\mu + tZ\right)Z\mathbb{I}\{\beta_N^\top\mu \in [\mu_l, \mu_r], Z \in [q_{1-\alpha}(Z), 2q_{1-\alpha}(Z)]\}|Z \geq q_{1-\alpha}(Z)\right].$$

Note that when
$$\beta_N^\top\mu \in [\mu_l, \mu_r], t \in [t_l, t_r], Z \in [q_{1-\alpha}(Z), 2q_{1-\alpha}(Z)],$$

we have
$$\phi'\left(\beta_N^\top\mu + tZ\right) > c_0.$$

Then, $h'(t)$ is lower bounded by
$$h'(t)$$
$$\geq c_0\mathbb{E}\left[Z\mathbb{I}\{\beta_N^\top\mu \in [\mu_l, \mu_r], Z \in [q_{1-\alpha}(Z), 2q_{1-\alpha}(Z)]\}|Z \geq q_{1-\alpha}(Z)\right]$$
$$\geq c_0 c_1\mathbb{E}\left[Z\mathbb{I}\{Z \in [q_{1-\alpha}(Z), 2q_{1-\alpha}(Z)]\}|Z \geq q_{1-\alpha}(Z)\right].$$

Since $\alpha \leq 0.2$, we have

$$\mathbb{E}\left[Z\mathbb{I}\{Z \in [q_{1-\alpha}(Z), 2q_{1-\alpha}(Z)]\}|Z \geq q_{1-\alpha}(Z)\right]$$
$$\geq \quad \frac{1}{2}\mathbb{E}\left[Z|Z \geq q_{1-\alpha}(Z)\right],$$

which yields the desired lower bound. $\qquad\square$

## B NUMERICAL RESULTS

### B.1 SYNTHETIC DATA

In this section, we report additional results on synthetic dataset. We plot average calibration errors, average ECE and average MCE for $\alpha \in [2\%, 10\%]$ in Figure 3. MCE is defined as

$$\text{MCE} \triangleq \max_{m \in \{1,\ldots,M\}} \left| \frac{\sum_{k \in B_m} y_k}{|B_m|} - \frac{\sum_{k \in B_m} f(x_k)}{|B_m|} \right|, \tag{7}$$

.We test methods with different hyperparameters. Specifically, we test our method with $S = 3$, and with identity mapping $\phi(x) = x$ (denoted as VAD(p)). The results are summarized in Tables 4, 5 and 6. We find that for the VAD method, $S = 3$ outperforms $S = 2$, with the cost of more training resources needed. VAD(p) and VAD have similar performances.

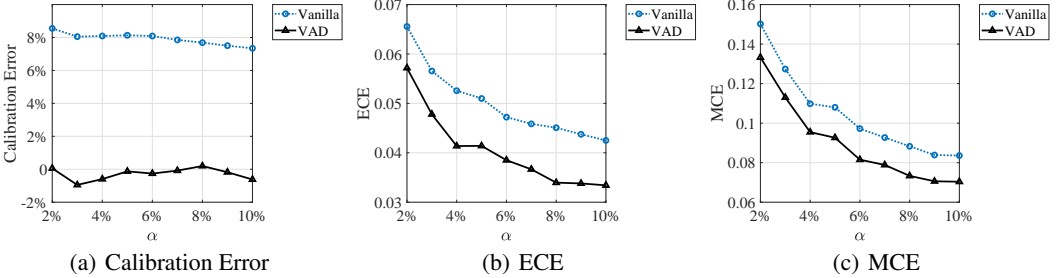

(a) Calibration Error      (b) ECE      (c) MCE

Figure 3: Average calibration errors, ECE, and MCE on the synthetic data

Tables 7 reports the Log Loss reduction on the selection set. Log Loss is defined as

$$\text{LogLoss}(\tilde{\mathcal{D}}_{\text{test}}^{\alpha}, f) = -\frac{1}{|\tilde{\mathcal{D}}_{\text{test}}^{\alpha}|} \sum_{i \in \tilde{\mathcal{D}}_{\text{test}}^{\alpha}} y_i \log(f(x_i)) + (1 - y_i) \log(1 - f(x_i)).$$

Then, the Log Loss reduction is defined by

$$\frac{\text{LogLoss}(\tilde{\mathcal{D}}_{\text{test}}^{\alpha}, f_{\text{VAD}})}{\text{LogLoss}(\tilde{\mathcal{D}}_{\text{test}}^{\alpha}, f_{\text{vanilla}})} - 1.$$

Negative Log Loss reduction means that we achieve lower loss. We find that after applying our method, we achieve lower Log Loss, meaning that we improve the prediction quality.

Table 6: Average and standard errors of MCE on synthetic data

| $\alpha$ | Vanilla | VAD | VAD(p) | VAD (S=3) |
|---|---|---|---|---|
| $\alpha$=2% | 0.1502±0.0041 | 0.1333±0.0035 | 0.1327±0.0035 | **0.1300**±0.0034 |
| $\alpha$=3% | 0.1274±0.0041 | 0.1130±0.0031 | **0.1118**±0.0032 | 0.1119±0.0031 |
| $\alpha$=4% | 0.1098±0.0033 | 0.0955±0.0028 | **0.0944**±0.0028 | 0.0965±0.0028 |
| $\alpha$=5% | 0.1080±0.0033 | 0.0926±0.0026 | **0.0915**±0.0026 | 0.0916±0.0025 |
| $\alpha$=6% | 0.0972±0.0030 | 0.0815±0.0025 | 0.0818±0.0024 | **0.0810**±0.0024 |
| $\alpha$=7% | 0.0927±0.0029 | 0.0789±0.0025 | **0.0784**±0.0024 | 0.0793±0.0024 |
| $\alpha$=8% | 0.0883±0.0028 | 0.0733±0.0023 | **0.0731**±0.0023 | 0.0743±0.0024 |
| $\alpha$=9% | 0.0839±0.0030 | 0.0706±0.0023 | 0.0699±0.0024 | **0.0698**±0.0023 |
| $\alpha$=10% | 0.0836±0.0027 | 0.0704±0.0022 | 0.0698±0.0022 | **0.0690**±0.0021 |

Table 4: Average and standard errors of calibration errors on synthetic data

| $\alpha$ | Vanilla | VAD | VAD(p) | VAD (S=3) |
|---|---|---|---|---|
| $\alpha$=2% | 8.55%±0.68% | **0.06%**±0.72% | 1.00%±0.70% | -0.34%±0.63% |
| $\alpha$=3% | 8.05%±0.70% | -0.95%±0.69% | **0.28%**±0.68% | -0.58%±0.64% |
| $\alpha$=4% | 8.09%±0.70% | -0.60%±0.67% | 0.40%±0.65% | **-0.23%**±0.67% |
| $\alpha$=5% | 8.13%±0.69% | -0.13%±0.71% | 0.95%±0.69% | **-0.10%**±0.66% |
| $\alpha$=6% | 8.09%±0.69% | -0.26%±0.71% | 0.75%±0.71% | **-0.10%**±0.67% |
| $\alpha$=7% | 7.85%±0.71% | **-0.08%**±0.73% | 1.11%±0.71% | -0.13%±0.70% |
| $\alpha$=8% | 7.69%±0.72% | **0.20%**±0.66% | 1.12%±0.66% | -0.24%±0.67% |
| $\alpha$=9% | 7.50%±0.74% | -0.18%±0.71% | 0.94%±0.71% | **-0.10%**±0.71% |
| $\alpha$=10% | 7.34%±0.75% | -0.62%±0.73% | 0.60%±0.72% | **-0.08%**±0.70% |

Table 5: Average and standard errors of ECE on synthetic data

| $\alpha$ | Vanilla | VAD | VAD(p) | VAD (S=3) |
|---|---|---|---|---|
| $\alpha$=2% | 0.0656±0.0019 | 0.0572±0.0013 | 0.0566±0.0013 | **0.0555**±0.0013 |
| $\alpha$=3% | 0.0566±0.0020 | 0.0478±0.0013 | 0.0474±0.0013 | **0.0463**±0.0013 |
| $\alpha$=4% | 0.0526±0.0021 | 0.0414±0.0014 | **0.0409**±0.0013 | 0.0416±0.0012 |
| $\alpha$=5% | 0.0510±0.0020 | 0.0414±0.0013 | 0.0410±0.0013 | **0.0404**±0.0012 |
| $\alpha$=6% | 0.0472±0.0021 | 0.0385±0.0014 | 0.0384±0.0013 | **0.0369**±0.0013 |
| $\alpha$=7% | 0.0459±0.0021 | 0.0367±0.0014 | 0.0363±0.0014 | **0.0358**±0.0013 |
| $\alpha$=8% | 0.0451±0.0020 | 0.0340±0.0012 | 0.0341±0.0013 | **0.0339**±0.0013 |
| $\alpha$=9% | 0.0437±0.0021 | **0.0338**±0.0014 | **0.0338**±0.0014 | 0.0338±0.0014 |
| $\alpha$=10% | 0.0425±0.0021 | 0.0334±0.0015 | 0.0331±0.0014 | **0.0326**±0.0013 |

Table 7: Log loss reduction on synthetic data by applying VAD

| $\alpha$ | 2% | 3% | 4% | 5% | 6% | 7% | 8% | 9% | 10% |
|---|---|---|---|---|---|---|---|---|---|
| | -0.45% | -0.42% | -0.42% | -0.37% | -0.35% | -0.33% | -0.37% | -0.32% | -0.30% |

## B.2 CRITEO AD KAGGLE DATASET

In this section, we report additional results on the Criteo Ad Kaggle dataset. Similarly with Appendix B.1, we test more hyperparameters. Note that for the VAD method with $S = 3$, we replicate 26 times. The results are summarized in Tables 8, 9 and 10. We plot average MCE for $\alpha \in [2\%, 10\%]$ in Figure 4. We find that for the VAD method, $S = 3$ outperforms $S = 2$ slightly, with the cost of more training resources needed. VAD(p) and VAD have similar performances. In addition, Table 11 reports the Log Loss reduction. We find that after applying our method, we achieve lower Log Loss uniformly, meaning that we improve the prediction quality.

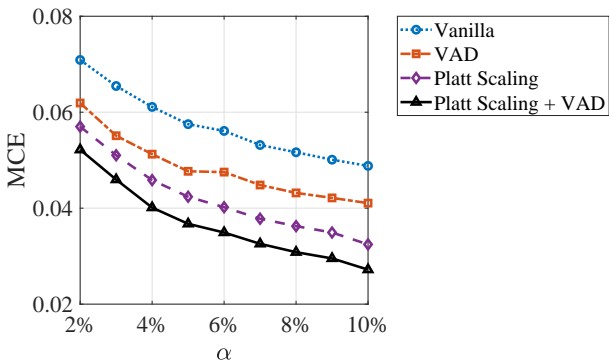

Figure 4: MCE the Criteo Ad Kaggle dataset

Table 8: Average and standard errors of calibration errors on Criteo Ad Kaggle dataset

| $\alpha$ | Methods | Original | VAD+ | VAD(p)+ | VAD+ (S=3) |
|---|---|---|---|---|---|
| 3% | Vanilla | 3.53%±0.45% | **-0.38**%±0.50% | -0.54%±0.46% | -0.72%±0.61% |
| | Histogram Binning | 2.19%±0.06% | 1.40%±0.06% | 1.23%±0.06% | 1.40%±0.07% |
| | Platt Scaling | 1.77%±0.12% | 0.98%±0.11% | 0.81%±0.11% | 1.00%±0.13% |
| | Scaling-Binning | 2.27%±0.12% | 1.48%±0.11% | 1.30%±0.11% | 1.50%±0.14% |
| | Isotonic Regression | 1.86%±0.05% | 1.06%±0.05% | 0.89%±0.06% | 1.06%±0.05% |
| 5% | Vanilla | 3.82%±0.52% | -0.36%±0.56% | **-0.14%**±0.53% | -0.76%±0.69% |
| | Histogram Binning | 2.03%±0.06% | 1.19%±0.05% | 1.10%±0.06% | 1.18%±0.06% |
| | Platt Scaling | 1.88%±0.10% | 1.03%±0.10% | 0.95%±0.10% | 1.06%±0.12% |
| | Scaling-Binning | 2.22%±0.10% | 1.37%±0.10% | 1.28%±0.10% | 1.39%±0.12% |
| | Isotonic Regression | 1.79%±0.05% | 0.94%±0.05% | 0.86%±0.05% | 0.95%±0.05% |
| 7% | Vanilla | 3.90%±0.57% | -0.42%±0.61% | **0.04**%±0.57% | -0.90%±0.74% |
| | Histogram Binning | 1.92%±0.03% | 1.05%±0.04% | 1.02%±0.04% | 1.03%±0.05% |
| | Platt Scaling | 1.86%±0.08% | 0.99%±0.08% | 0.95%±0.08% | 0.97%±0.10% |
| | Scaling-Binning | 2.13%±0.09% | 1.26%±0.08% | 1.22%±0.08% | 1.24%±0.10% |
| | Isotonic Regression | 1.72%±0.04% | 0.84%±0.04% | 0.81%±0.04% | 0.81%±0.05% |
| 9% | Vanilla | 3.93%±0.61% | -0.46%±0.64% | **0.16**%±0.61% | -1.00%±0.79% |
| | Histogram Binning | 1.89%±0.03% | 1.00%±0.04% | 1.01%±0.04% | 0.95%±0.05% |
| | Platt Scaling | 1.82%±0.08% | 0.93%±0.07% | 0.93%±0.08% | 0.89%±0.10% |
| | Scaling-Binning | 2.04%±0.08% | 1.15%±0.07% | 1.15%±0.08% | 1.11%±0.10% |
| | Isotonic Regression | 1.74%±0.03% | 0.84%±0.03% | 0.85%±0.04% | 0.79%±0.05% |

Table 9: Average and standard errors of ECE on Criteo Ad Kaggle dataset

| $\alpha$ | Methods | Original | VAD+ | VAD(p)+ | VAD+ (S=3) |
|---|---|---|---|---|---|
| 3% | Vanilla | 0.0290±0.0026 | 0.0218±0.0018 | 0.0219±0.0015 | 0.0223±0.0020 |
| | Histogram Binning | 0.0190±0.0003 | 0.0160±0.0003 | 0.0154±0.0003 | 0.0159±0.0003 |
| | Platt Scaling | 0.0169±0.0005 | 0.0144±0.0004 | **0.0142**±0.0003 | 0.0145±0.0004 |
| | Scaling-Binning | 0.0192±0.0006 | 0.0161±0.0005 | 0.0153±0.0004 | 0.0160±0.0006 |
| | Isotonic Regression | 0.0181±0.0003 | 0.0157±0.0003 | 0.0155±0.0003 | 0.0155±0.0003 |
| 5% | Vanilla | 0.0287±0.0028 | 0.0213±0.0020 | 0.0211±0.0017 | 0.0217±0.0024 |
| | Histogram Binning | 0.0163±0.0003 | 0.0133±0.0002 | 0.0128±0.0002 | 0.0132±0.0003 |
| | Platt Scaling | 0.0151±0.0005 | 0.0121±0.0003 | 0.0120±0.0003 | **0.0119**±0.0004 |
| | Scaling-Binning | 0.0166±0.0005 | 0.0132±0.0004 | 0.0128±0.0004 | 0.0133±0.0005 |
| | Isotonic Regression | 0.0156±0.0003 | 0.0130±0.0003 | 0.0129±0.0003 | 0.0130±0.0003 |
| 7% | Vanilla | 0.0281±0.0029 | 0.0211±0.0021 | 0.0208±0.0019 | 0.0216±0.0025 |
| | Histogram Binning | 0.0146±0.0002 | 0.0117±0.0002 | 0.0114±0.0002 | 0.0117±0.0003 |
| | Platt Scaling | 0.0137±0.0004 | 0.0108±0.0003 | **0.0107**±0.0003 | **0.0107**±0.0003 |
| | Scaling-Binning | 0.0149±0.0004 | 0.0116±0.0003 | 0.0114±0.0003 | 0.0118±0.0003 |
| | Isotonic Regression | 0.0138±0.0002 | 0.0114±0.0002 | 0.0113±0.0002 | 0.0112±0.0003 |
| 9% | Vanilla | 0.0274±0.0029 | 0.0208±0.0022 | 0.0205±0.0020 | 0.0212±0.0026 |
| | Histogram Binning | 0.0135±0.0002 | 0.0107±0.0002 | 0.0105±0.0002 | 0.0106±0.0002 |
| | Platt Scaling | 0.0125±0.0004 | 0.0096±0.0002 | 0.0096±0.0003 | **0.0095**±0.0003 |
| | Scaling-Binning | 0.0136±0.0004 | 0.0104±0.0002 | 0.0103±0.0003 | 0.0105±0.0003 |
| | Isotonic Regression | 0.0129±0.0002 | 0.0104±0.0002 | 0.0104±0.0002 | 0.0104±0.0003 |

Table 10: Average and standard errors of MCE on Criteo Ad Kaggle dataset

| $\alpha$ | Methods | Original | VAD+ | VAD(p)+ | VAD+ (S=3) |
|---|---|---|---|---|---|
| 3% | Vanilla | 0.0655±0.0037 | 0.0551±0.0028 | 0.0561±0.0026 | 0.0547±0.0035 |
| | Histogram Binning | 0.0524±0.0011 | 0.0467±0.0011 | 0.0459±0.0011 | **0.0448**±0.0012 |
| | Platt Scaling | 0.0510±0.0015 | 0.0460±0.0013 | 0.0457±0.0013 | 0.0454±0.0017 |
| | Scaling-Binning | 0.0513±0.0013 | 0.0464±0.0012 | 0.0451±0.0011 | 0.0462±0.0013 |
| | Isotonic Regression | 0.0572±0.0016 | 0.0515±0.0015 | 0.0510±0.0016 | 0.0510±0.0018 |
| 5% | Vanilla | 0.0575±0.0035 | 0.0477±0.0027 | 0.0486±0.0024 | 0.0469±0.0034 |
| | Histogram Binning | 0.0475±0.0011 | 0.0420±0.0011 | 0.0412±0.0012 | 0.0417±0.0012 |
| | Platt Scaling | 0.0424±0.0010 | 0.0367±0.0010 | 0.0367±0.0010 | **0.0363**±0.0012 |
| | Scaling-Binning | 0.0451±0.0013 | 0.0398±0.0013 | 0.0388±0.0013 | 0.0396±0.0016 |
| | Isotonic Regression | 0.0474±0.0016 | 0.0416±0.0015 | 0.0415±0.0015 | 0.0404±0.0012 |
| 7% | Vanilla | 0.0532±0.0035 | 0.0448±0.0026 | 0.0460±0.0024 | 0.0451±0.0031 |
| | Histogram Binning | 0.0402±0.0009 | 0.0351±0.0009 | 0.0344±0.0010 | 0.0344±0.0012 |
| | Platt Scaling | 0.0378±0.0010 | 0.0326±0.0009 | 0.0327±0.0010 | **0.0325**±0.0012 |
| | Scaling-Binning | 0.0405±0.0011 | 0.0354±0.0011 | 0.0339±0.0010 | 0.0361±0.0016 |
| | Isotonic Regression | 0.0420±0.0012 | 0.0366±0.0012 | 0.0364±0.0011 | 0.0357±0.0011 |
| 9% | Vanilla | 0.0501±0.0032 | 0.0421±0.0024 | 0.0432±0.0022 | 0.0422±0.0029 |
| | Histogram Binning | 0.0363±0.0008 | 0.0310±0.0007 | 0.0309±0.0008 | 0.0296±0.0008 |
| | Platt Scaling | 0.0349±0.0008 | **0.0295**±0.0008 | 0.0296±0.0008 | 0.0296±0.0009 |
| | Scaling-Binning | 0.0376±0.0010 | 0.0327±0.0009 | 0.0312±0.0009 | 0.0333±0.0012 |
| | Isotonic Regression | 0.0386±0.0010 | 0.0335±0.0010 | 0.0332±0.0009 | 0.0322±0.0009 |

Table 11: Log loss reduction on Criteo Ad Kaggle dataset by applying VAD

| Methods | $\alpha$=2% | $\alpha$=3% | $\alpha$=4% | $\alpha$=5% | $\alpha$=6% | $\alpha$=7% | $\alpha$=8% | $\alpha$=9% | $\alpha$=10% |
|---|---|---|---|---|---|---|---|---|---|
| Vanilla | -0.27% | -0.26% | -0.25% | -0.23% | -0.21% | -0.20% | -0.19% | -0.18% | -0.17% |
| Histogram Binning | -0.08% | -0.07% | -0.06% | -0.05% | -0.04% | -0.04% | -0.04% | -0.03% | -0.03% |
| Platt Scaling | -0.05% | -0.05% | -0.05% | -0.04% | -0.04% | -0.04% | -0.03% | -0.03% | -0.03% |
| Scaling-Binning | -0.09% | -0.07% | -0.06% | -0.06% | -0.05% | -0.04% | -0.04% | -0.04% | -0.04% |
| Isotonic Regression | -0.05% | -0.05% | -0.04% | -0.04% | -0.04% | -0.03% | -0.03% | -0.03% | -0.03% |

We further check the performance using different bin numbers $M \in \{30, 40, 60, 70\}$ in Tables 12 to 15. We observe that regardless of the number of bins we choose, our method always outperforms.

Table 12: Average and standard errors of ECE on Criteo Ad Kaggle dataset ($M = 30, 40$)

| | | Original (M=30) | VAD+ (M=30) | Original (M=40) | VAD+ (M=40) |
|---|---|---|---|---|---|
| 3% | Vanilla | 0.0279±0.0027 | 0.0204±0.0019 | 0.0285±0.0026 | 0.0211±0.0018 |
| | Histogram Binning | 0.0174±0.0003 | 0.0139±0.0003 | 0.0181±0.0003 | 0.0149±0.0003 |
| | Platt Scaling | 0.0152±0.0006 | 0.0122±0.0004 | 0.0159±0.0005 | 0.0132±0.0004 |
| | Scaling-Binning | 0.0179±0.0006 | 0.0143±0.0005 | 0.0185±0.0006 | 0.0152±0.0005 |
| | Isotonic Regression | 0.0161±0.0003 | 0.0133±0.0003 | 0.0171±0.0003 | 0.0145±0.0003 |
| 5% | Vanilla | 0.0283±0.0029 | 0.0203±0.0021 | 0.0285±0.0029 | 0.0209±0.0020 |
| | Histogram Binning | 0.0150±0.0003 | 0.0115±0.0003 | 0.0158±0.0003 | 0.0126±0.0002 |
| | Platt Scaling | 0.0138±0.0005 | 0.0103±0.0003 | 0.0144±0.0005 | 0.0112±0.0003 |
| | Scaling-Binning | 0.0156±0.0006 | 0.0117±0.0004 | 0.0163±0.0005 | 0.0127±0.0004 |
| | Isotonic Regression | 0.0143±0.0003 | 0.0114±0.0003 | 0.0149±0.0003 | 0.0122±0.0002 |
| 7% | Vanilla | 0.0276±0.0030 | 0.0203±0.0022 | 0.0277±0.0030 | 0.0207±0.0021 |
| | Histogram Binning | 0.0136±0.0002 | 0.0103±0.0002 | 0.0140±0.0002 | 0.0111±0.0002 |
| | Platt Scaling | 0.0127±0.0005 | 0.0092±0.0003 | 0.0132±0.0004 | 0.0100±0.0003 |
| | Scaling-Binning | 0.0142±0.0004 | 0.0104±0.0003 | 0.0145±0.0004 | 0.0110±0.0003 |
| | Isotonic Regression | 0.0128±0.0002 | 0.0101±0.0002 | 0.0132±0.0002 | 0.0106±0.0002 |
| 9% | Vanilla | 0.0269±0.0030 | 0.0202±0.0022 | 0.0272±0.0030 | 0.0205±0.0022 |
| | Histogram Binning | 0.0126±0.0002 | 0.0094±0.0002 | 0.0129±0.0002 | 0.0099±0.0002 |
| | Platt Scaling | 0.0117±0.0004 | 0.0083±0.0003 | 0.0121±0.0004 | 0.0090±0.0002 |
| | Scaling-Binning | 0.0128±0.0004 | 0.0092±0.0003 | 0.0131±0.0004 | 0.0096±0.0003 |
| | Isotonic Regression | 0.0119±0.0002 | 0.0090±0.0002 | 0.0124±0.0002 | 0.0098±0.0002 |

Table 13: Average and standard errors of ECE on Criteo Ad Kaggle dataset ($M = 60, 70$)

| | | Original (M=60) | VAD+ (M=60) | Original (M=70) | VAD+ (M=70) |
|---|---|---|---|---|---|
| 3% | Vanilla | 0.0293±0.0026 | 0.0226±0.0017 | 0.0300±0.0025 | 0.0232±0.0017 |
| | Histogram Binning | 0.0196±0.0003 | 0.0168±0.0003 | 0.0204±0.0003 | 0.0177±0.0003 |
| | Platt Scaling | 0.0177±0.0005 | 0.0154±0.0003 | 0.0185±0.0005 | 0.0163±0.0003 |
| | Scaling-Binning | 0.0199±0.0006 | 0.0169±0.0004 | 0.0206±0.0005 | 0.0177±0.0004 |
| | Isotonic Regression | 0.0188±0.0003 | 0.0166±0.0003 | 0.0195±0.0003 | 0.0173±0.0003 |
| 5% | Vanilla | 0.0292±0.0028 | 0.0219±0.0019 | 0.0295±0.0028 | 0.0224±0.0019 |
| | Histogram Binning | 0.0169±0.0003 | 0.0140±0.0002 | 0.0174±0.0003 | 0.0147±0.0002 |
| | Platt Scaling | 0.0158±0.0004 | 0.0131±0.0003 | 0.0163±0.0004 | 0.0138±0.0003 |
| | Scaling-Binning | 0.0172±0.0005 | 0.0140±0.0004 | 0.0176±0.0005 | 0.0146±0.0003 |
| | Isotonic Regression | 0.0163±0.0002 | 0.0138±0.0002 | 0.0171±0.0002 | 0.0149±0.0002 |
| 7% | Vanilla | 0.0283±0.0029 | 0.0214±0.0020 | 0.0286±0.0029 | 0.0218±0.0020 |
| | Histogram Binning | 0.0151±0.0002 | 0.0124±0.0002 | 0.0157±0.0002 | 0.0131±0.0002 |
| | Platt Scaling | 0.0141±0.0004 | 0.0112±0.0002 | 0.0147±0.0004 | 0.0120±0.0002 |
| | Scaling-Binning | 0.0155±0.0004 | 0.0125±0.0002 | 0.0159±0.0004 | 0.0130±0.0003 |
| | Isotonic Regression | 0.0146±0.0002 | 0.0123±0.0002 | 0.0149±0.0002 | 0.0127±0.0002 |
| 9% | Vanilla | 0.0276±0.0029 | 0.0212±0.0021 | 0.0277±0.0029 | 0.0213±0.0021 |
| | Histogram Binning | 0.0139±0.0002 | 0.0113±0.0002 | 0.0142±0.0002 | 0.0116±0.0001 |
| | Platt Scaling | 0.0129±0.0003 | 0.0102±0.0002 | 0.0133±0.0003 | 0.0106±0.0002 |
| | Scaling-Binning | 0.0139±0.0004 | 0.0109±0.0002 | 0.0143±0.0003 | 0.0114±0.0002 |
| | Isotonic Regression | 0.0134±0.0002 | 0.0110±0.0002 | 0.0138±0.0002 | 0.0115±0.0002 |

Table 14: Average and standard errors of MCE on Criteo Ad Kaggle dataset ($M = 30, 40$)

|  |  | Original (M=30) | VAD+ (M=30) | Original (M=40) | VAD+ (M=40) |
|---|---|---|---|---|---|
| 3% | Vanilla | 0.0542±0.0035 | 0.0442±0.0026 | 0.0589±0.0033 | 0.0494±0.0024 |
|  | Histogram Binning | 0.0434±0.0010 | 0.0380±0.0010 | 0.0482±0.0011 | 0.0427±0.0011 |
|  | Platt Scaling | 0.0398±0.0011 | 0.0341±0.0011 | 0.0445±0.0013 | 0.0391±0.0012 |
|  | Scaling-Binning | 0.0415±0.0013 | 0.0364±0.0013 | 0.0483±0.0014 | 0.0431±0.0014 |
|  | Isotonic Regression | 0.0465±0.0014 | 0.0406±0.0013 | 0.0531±0.0015 | 0.0473±0.0014 |
| 5% | Vanilla | 0.0485±0.0035 | 0.0396±0.0025 | 0.0538±0.0037 | 0.0442±0.0024 |
|  | Histogram Binning | 0.0379±0.0010 | 0.0326±0.0010 | 0.0434±0.0010 | 0.0382±0.0010 |
|  | Platt Scaling | 0.0343±0.0009 | 0.0286±0.0008 | 0.0390±0.0010 | 0.0337±0.0009 |
|  | Scaling-Binning | 0.0369±0.0011 | 0.0317±0.0011 | 0.0403±0.0013 | 0.0351±0.0013 |
|  | Isotonic Regression | 0.0389±0.0012 | 0.0336±0.0011 | 0.0424±0.0012 | 0.0368±0.0011 |
| 7% | Vanilla | 0.0456±0.0033 | 0.0375±0.0025 | 0.0492±0.0034 | 0.0406±0.0024 |
|  | Histogram Binning | 0.0341±0.0007 | 0.0287±0.0007 | 0.0364±0.0008 | 0.0311±0.0008 |
|  | Platt Scaling | 0.0302±0.0009 | 0.0248±0.0008 | 0.0340±0.0009 | 0.0285±0.0009 |
|  | Scaling-Binning | 0.0340±0.0009 | 0.0290±0.0009 | 0.0367±0.0009 | 0.0316±0.0009 |
|  | Isotonic Regression | 0.0349±0.0009 | 0.0293±0.0010 | 0.0395±0.0009 | 0.0337±0.0009 |
| 9% | Vanilla | 0.0434±0.0031 | 0.0353±0.0024 | 0.0468±0.0033 | 0.0394±0.0025 |
|  | Histogram Binning | 0.0307±0.0006 | 0.0255±0.0006 | 0.0346±0.0008 | 0.0293±0.0008 |
|  | Platt Scaling | 0.0273±0.0007 | 0.0219±0.0007 | 0.0308±0.0008 | 0.0255±0.0007 |
|  | Scaling-Binning | 0.0316±0.0010 | 0.0267±0.0010 | 0.0338±0.0010 | 0.0288±0.0010 |
|  | Isotonic Regression | 0.0331±0.0009 | 0.0274±0.0009 | 0.0358±0.0008 | 0.0303±0.0008 |

Table 15: Average and standard errors of MCE on Criteo Ad Kaggle dataset ($M = 60, 70$)

|  |  | Original (M=60) | VAD+ (M=60) | Original (M=70) | VAD+ (M=70) |
|---|---|---|---|---|---|
| 3% | Vanilla | 0.0695±0.0034 | 0.0595±0.0027 | 0.0752±0.0035 | 0.0644±0.0028 |
|  | Histogram Binning | 0.0568±0.0012 | 0.0513±0.0012 | 0.0610±0.0012 | 0.0560±0.0012 |
|  | Platt Scaling | 0.0548±0.0012 | 0.0494±0.0011 | 0.0600±0.0012 | 0.0549±0.0011 |
|  | Scaling-Binning | 0.0550±0.0012 | 0.0500±0.0011 | 0.0598±0.0015 | 0.0552±0.0014 |
|  | Isotonic Regression | 0.0606±0.0014 | 0.0552±0.0013 | 0.0658±0.0016 | 0.0611±0.0015 |
| 5% | Vanilla | 0.0631±0.0035 | 0.0540±0.0028 | 0.0634±0.0033 | 0.0543±0.0024 |
|  | Histogram Binning | 0.0508±0.0011 | 0.0455±0.0011 | 0.0540±0.0011 | 0.0485±0.0010 |
|  | Platt Scaling | 0.0473±0.0010 | 0.0419±0.0010 | 0.0489±0.0010 | 0.0436±0.0010 |
|  | Scaling-Binning | 0.0484±0.0014 | 0.0433±0.0013 | 0.0511±0.0014 | 0.0461±0.0013 |
|  | Isotonic Regression | 0.0492±0.0015 | 0.0440±0.0014 | 0.0533±0.0015 | 0.0479±0.0014 |
| 7% | Vanilla | 0.0559±0.0032 | 0.0478±0.0024 | 0.0605±0.0032 | 0.0517±0.0026 |
|  | Histogram Binning | 0.0439±0.0010 | 0.0387±0.0009 | 0.0458±0.0009 | 0.0407±0.0010 |
|  | Platt Scaling | 0.0404±0.0011 | 0.0349±0.0010 | 0.0446±0.0008 | 0.0395±0.0007 |
|  | Scaling-Binning | 0.0442±0.0012 | 0.0393±0.0011 | 0.0462±0.0012 | 0.0411±0.0012 |
|  | Isotonic Regression | 0.0454±0.0010 | 0.0399±0.0010 | 0.0482±0.0011 | 0.0427±0.0011 |
| 9% | Vanilla | 0.0533±0.0032 | 0.0447±0.0023 | 0.0553±0.0032 | 0.0478±0.0026 |
|  | Histogram Binning | 0.0398±0.0009 | 0.0345±0.0009 | 0.0424±0.0010 | 0.0372±0.0010 |
|  | Platt Scaling | 0.0372±0.0008 | 0.0318±0.0007 | 0.0397±0.0010 | 0.0346±0.0009 |
|  | Scaling-Binning | 0.0401±0.0010 | 0.0349±0.0009 | 0.0425±0.0010 | 0.0373±0.0009 |
|  | Isotonic Regression | 0.0416±0.0010 | 0.0364±0.0010 | 0.0452±0.0012 | 0.0399±0.0012 |

We then check the performance using different $S \in \{4, 5, 6\}$ in Tables 16 to 18. We find that different $S$s have similar performance, except that for the Vanilla method, VAD+Vanilla with $S = 6$ is indeed better.

Table 16: Average and standard errors of calibration errors on Criteo Ad Kaggle dataset ($S = 4, 5, 6$)

|  |  | Original | VAD+ (S=4) | VAD+ (S=5) | VAD+ (S=6) |
|---|---|---|---|---|---|
| 3% | Vanilla | 3.53%±0.45% | -0.88%±0.70% | -0.20%±0.76% | 0.40%±0.70% |
|  | Histogram Binning | 2.19%±0.06% | 1.42%±0.08% | 1.45%±0.09% | 1.46%±0.10% |
|  | Platt Scaling | 1.77%±0.12% | 1.12%±0.15% | 1.17%±0.17% | 1.12%±0.18% |
|  | Scaling-Binning | 2.27%±0.12% | 1.62%±0.15% | 1.68%±0.19% | 1.61%±0.19% |
|  | Isotonic Regression | 1.86%±0.05% | 1.08%±0.06% | 1.08%±0.07% | 1.12%±0.08% |
| 5% | Vanilla | 3.82%±0.52% | -0.96%±0.77% | -0.21%±0.85% | 0.39%±0.79% |
|  | Histogram Binning | 2.03%±0.06% | 1.17%±0.06% | 1.21%±0.07% | 1.15%±0.07% |
|  | Platt Scaling | 1.88%±0.10% | 1.16%±0.13% | 1.24%±0.15% | 1.16%±0.15% |
|  | Scaling-Binning | 2.22%±0.10% | 1.49%±0.13% | 1.57%±0.15% | 1.47%±0.16% |
|  | Isotonic Regression | 1.79%±0.05% | 0.91%±0.06% | 0.95%±0.07% | 0.89%±0.07% |
| 7% | Vanilla | 3.90%±0.57% | -1.13%±0.82% | -0.37%±0.91% | 0.27%±0.85% |
|  | Histogram Binning | 1.92%±0.03% | 0.99%±0.05% | 1.01%±0.06% | 0.98%±0.06% |
|  | Platt Scaling | 1.86%±0.08% | 1.04%±0.11% | 1.11%±0.12% | 1.06%±0.13% |
|  | Scaling-Binning | 2.13%±0.09% | 1.31%±0.11% | 1.38%±0.12% | 1.32%±0.13% |
|  | Isotonic Regression | 1.72%±0.04% | 0.77%±0.06% | 0.79%±0.07% | 0.77%±0.07% |
| 9% | Vanilla | 3.93%±0.61% | -1.26%±0.85% | -0.48%±0.96% | 0.17%±0.91% |
|  | Histogram Binning | 1.89%±0.03% | 0.91%±0.05% | 0.95%±0.06% | 0.96%±0.05% |
|  | Platt Scaling | 1.82%±0.08% | 0.94%±0.10% | 1.01%±0.11% | 0.97%±0.12% |
|  | Scaling-Binning | 2.04%±0.08% | 1.17%±0.10% | 1.24%±0.12% | 1.20%±0.12% |
|  | Isotonic Regression | 1.74%±0.03% | 0.75%±0.05% | 0.80%±0.06% | 0.79%±0.05% |

Table 17: Average and standard errors of ECE on Criteo Ad Kaggle dataset ($S = 4, 5, 6$)

|  |  | Original | VAD+ (S=4) | VAD+ (S=5) | VAD+ (S=6) |
|---|---|---|---|---|---|
| 3% | Vanilla | 0.0290±0.0026 | 0.0224±0.0025 | 0.0214±0.0027 | 0.0193±0.0021 |
|  | Histogram Binning | 0.0190±0.0003 | 0.0161±0.0004 | 0.0162±0.0005 | 0.0161±0.0005 |
|  | Platt Scaling | 0.0169±0.0005 | 0.0147±0.0005 | 0.0151±0.0006 | 0.0148±0.0005 |
|  | Scaling-Binning | 0.0192±0.0006 | 0.0165±0.0007 | 0.0165±0.0008 | 0.0159±0.0008 |
|  | Isotonic Regression | 0.0181±0.0003 | 0.0152±0.0003 | 0.0153±0.0004 | 0.0151±0.0004 |
| 5% | Vanilla | 0.0287±0.0028 | 0.0213±0.0028 | 0.0203±0.0031 | 0.0184±0.0025 |
|  | Histogram Binning | 0.0163±0.0003 | 0.0133±0.0003 | 0.0132±0.0003 | 0.0130±0.0004 |
|  | Platt Scaling | 0.0151±0.0005 | 0.0120±0.0005 | 0.0124±0.0005 | 0.0119±0.0006 |
|  | Scaling-Binning | 0.0166±0.0005 | 0.0137±0.0006 | 0.0138±0.0007 | 0.0132±0.0008 |
|  | Isotonic Regression | 0.0156±0.0003 | 0.0130±0.0004 | 0.0131±0.0004 | 0.0130±0.0005 |
| 7% | Vanilla | 0.0281±0.0029 | 0.0213±0.0028 | 0.0202±0.0032 | 0.0181±0.0026 |
|  | Histogram Binning | 0.0146±0.0002 | 0.0117±0.0003 | 0.0114±0.0003 | 0.0113±0.0003 |
|  | Platt Scaling | 0.0137±0.0004 | 0.0108±0.0004 | 0.0112±0.0005 | 0.0108±0.0005 |
|  | Scaling-Binning | 0.0149±0.0004 | 0.0120±0.0004 | 0.0121±0.0005 | 0.0118±0.0005 |
|  | Isotonic Regression | 0.0138±0.0002 | 0.0113±0.0003 | 0.0112±0.0004 | 0.0110±0.0004 |
| 9% | Vanilla | 0.0274±0.0029 | 0.0208±0.0029 | 0.0198±0.0033 | 0.0177±0.0027 |
|  | Histogram Binning | 0.0135±0.0002 | 0.0105±0.0002 | 0.0105±0.0003 | 0.0103±0.0003 |
|  | Platt Scaling | 0.0125±0.0004 | 0.0095±0.0004 | 0.0099±0.0004 | 0.0094±0.0004 |
|  | Scaling-Binning | 0.0136±0.0004 | 0.0107±0.0004 | 0.0109±0.0004 | 0.0105±0.0004 |
|  | Isotonic Regression | 0.0129±0.0002 | 0.0105±0.0003 | 0.0103±0.0003 | 0.0101±0.0004 |

Table 18: Average and standard errors of MCE on Criteo Ad Kaggle dataset ($S = 4, 5, 6$)

|     |                     | Original            | VAD+ (S=4)          | VAD+ (S=5)          | VAD+ (S=6)          |
| --- | ------------------- | ------------------- | ------------------- | ------------------- | ------------------- |
| 3%  | Vanilla             | 0.0655±0.0037       | 0.0541±0.0039       | 0.0538±0.0045       | 0.0511±0.0037       |
|     | Histogram Binning   | 0.0524±0.0011       | 0.0457±0.0012       | 0.0453±0.0015       | 0.0460±0.0017       |
|     | Platt Scaling       | 0.0510±0.0015       | 0.0452±0.0019       | 0.0463±0.0021       | 0.0451±0.0019       |
|     | Scaling-Binning     | 0.0513±0.0013       | 0.0472±0.0012       | 0.0469±0.0013       | 0.0460±0.0011       |
|     | Isotonic Regression | 0.0572±0.0016       | 0.0495±0.0022       | 0.0500±0.0027       | 0.0488±0.0027       |
| 5%  | Vanilla             | 0.0575±0.0035       | 0.0455±0.0035       | 0.0442±0.0040       | 0.0413±0.0031       |
|     | Histogram Binning   | 0.0475±0.0011       | 0.0417±0.0014       | 0.0420±0.0016       | 0.0419±0.0018       |
|     | Platt Scaling       | 0.0424±0.0010       | 0.0360±0.0014       | 0.0363±0.0016       | 0.0352±0.0017       |
|     | Scaling-Binning     | 0.0451±0.0013       | 0.0392±0.0017       | 0.0390±0.0020       | 0.0388±0.0024       |
|     | Isotonic Regression | 0.0474±0.0016       | 0.0403±0.0015       | 0.0412±0.0017       | 0.0407±0.0017       |
| 7%  | Vanilla             | 0.0532±0.0035       | 0.0446±0.0034       | 0.0441±0.0039       | 0.0418±0.0034       |
|     | Histogram Binning   | 0.0402±0.0009       | 0.0347±0.0014       | 0.0331±0.0014       | 0.0333±0.0016       |
|     | Platt Scaling       | 0.0378±0.0010       | 0.0328±0.0015       | 0.0332±0.0018       | 0.0329±0.0022       |
|     | Scaling-Binning     | 0.0405±0.0011       | 0.0370±0.0017       | 0.0357±0.0014       | 0.0349±0.0016       |
|     | Isotonic Regression | 0.0420±0.0012       | 0.0359±0.0013       | 0.0357±0.0015       | 0.0355±0.0018       |
| 9%  | Vanilla             | 0.0501±0.0032       | 0.0418±0.0031       | 0.0417±0.0037       | 0.0401±0.0033       |
|     | Histogram Binning   | 0.0363±0.0008       | 0.0289±0.0008       | 0.0291±0.0010       | 0.0286±0.0011       |
|     | Platt Scaling       | 0.0349±0.0008       | 0.0304±0.0011       | 0.0309±0.0011       | 0.0307±0.0013       |
|     | Scaling-Binning     | 0.0376±0.0010       | 0.0338±0.0013       | 0.0336±0.0016       | 0.0319±0.0014       |
|     | Isotonic Regression | 0.0386±0.0010       | 0.0322±0.0012       | 0.0316±0.0014       | 0.0317±0.0016       |

Finally, we present histograms of the last-layer neuron (logit) in the neural networks to justify our Gaussianity assumptions in Theorem 4.1. Figure 5 plots the histograms of model predictions from models using three different random seeds, where the black line is the estimated Gaussian density. By the Kolmogorov-Smirnov test, we cannot reject the null hypothesis that the empirical distribution is Gaussian at the 5% significance level.

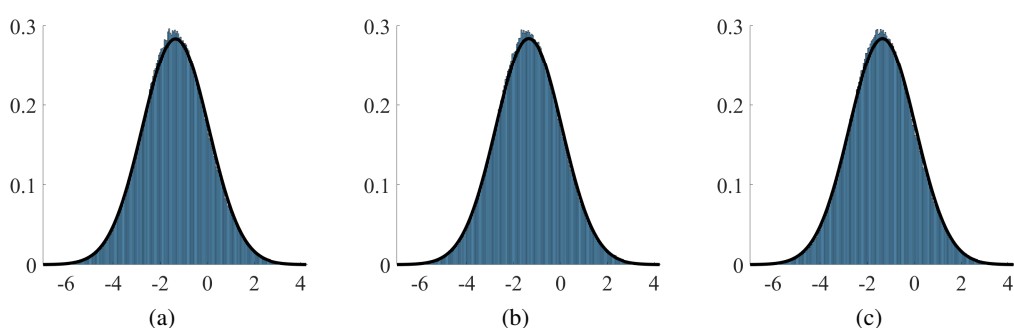

Figure 5: Histograms of the last-layer neuron (logit) of the neural networks

### B.3  AVAZU DATASET

**Dataset** We use the Avazu CTR Prediction dataset [4] to demonstrate our method's performance. The Avazu CTR Prediction dataset is a common benchmark dataset for CTR predictions. It consists 10 days of click-through data, ordered chronologically. Due to computational constraints in our experiments, we use the first 10 million samples, shuffle the dataset randomly, and split the whole dataset into 77% train $\mathcal{D}_{\text{train}}$, 1.5% validation-train $\mathcal{D}_{\text{val}-\text{train}}$, 1.5% validation-test $\mathcal{D}_{\text{val}-\text{test}}$, and 20% test $\mathcal{D}_{\text{test}}$ datasets.

**Base Model** We use the xDeepFM model (Lian et al., 2018) open-sourced in (Shen, 2017).

**Covariate Shift** We train another xDeepFM model. We randomly keep each data point in the original test set with probability $1 - p$ if $p < 0.2$, probability $0.1$ if $p > 0.3$ and probability $2.2 - 7 * p$ if $0.2 < p < 0.3$, where $p$ is the newly-trained xDeepFM model prediction for the data point. The training set remains the same. By doing this, we ensure that the distributional change is only a covariate shift, and the positive sample ratio in the test data is lower than the positive sample ratio in the training data, which is consistent with the real-world recommendation systems.

We report results for the calibration error, ECE, and MCE in Tables 19, 20, and 21, respectively. We observe that all the calibration methods in tandem with VAD achieve better performance than the sole calibration methods alone, which is consistent with results in our paper.

---

[4]https://www.kaggle.com/c/avazu-ctr-prediction

Table 19: Average and standard errors of calibration errors on the Avazu dataset

|     |                     | Original      | VAD+          |
|-----|---------------------|---------------|---------------|
| 3%  | Histogram Binning   | 3.65%±0.21%   | 1.32%±0.22%   |
|     | Platt Scaling       | 3.77%±0.24%   | 1.41%±0.24%   |
|     | Scaling-Binning     | 3.78%±0.24%   | 1.42%±0.24%   |
|     | Isotonic Regression | 3.77%±0.22%   | 1.42%±0.23%   |
| 5%  | Histogram Binning   | 5.10%±0.26%   | 2.76%±0.27%   |
|     | Platt Scaling       | 5.19%±0.27%   | 2.82%±0.27%   |
|     | Scaling-Binning     | 5.27%±0.27%   | 2.91%±0.27%   |
|     | Isotonic Regression | 5.17%±0.27%   | 2.80%±0.27%   |
| 7%  | Histogram Binning   | 5.79%±0.28%   | 3.47%±0.28%   |
|     | Platt Scaling       | 5.73%±0.28%   | 3.39%±0.28%   |
|     | Scaling-Binning     | 5.76%±0.28%   | 3.42%±0.28%   |
|     | Isotonic Regression | 5.67%±0.27%   | 3.32%±0.28%   |
| 9%  | Histogram Binning   | 6.07%±0.27%   | 3.78%±0.27%   |
|     | Platt Scaling       | 5.97%±0.29%   | 3.65%±0.29%   |
|     | Scaling-Binning     | 5.98%±0.29%   | 3.66%±0.29%   |
|     | Isotonic Regression | 5.91%±0.27%   | 3.59%±0.27%   |

Table 20: Average and standard errors of ECE on the Avazu dataset

|     |                     | Original         | VAD+             |
|-----|---------------------|------------------|------------------|
| 3%  | Histogram Binning   | 0.0209±0.0007    | 0.0186±0.0005    |
|     | Platt Scaling       | 0.0218±0.0007    | 0.0190±0.0007    |
|     | Scaling-Binning     | 0.0212±0.0006    | 0.0184±0.0006    |
|     | Isotonic Regression | 0.0215±0.0007    | 0.0189±0.0005    |
| 5%  | Histogram Binning   | 0.0208±0.0007    | 0.0173±0.0005    |
|     | Platt Scaling       | 0.0211±0.0008    | 0.0172±0.0007    |
|     | Scaling-Binning     | 0.0213±0.0008    | 0.0173±0.0007    |
|     | Isotonic Regression | 0.0210±0.0007    | 0.0171±0.0005    |
| 7%  | Histogram Binning   | 0.0202±0.0007    | 0.0163±0.0006    |
|     | Platt Scaling       | 0.0201±0.0008    | 0.0157±0.0007    |
|     | Scaling-Binning     | 0.0200±0.0008    | 0.0157±0.0007    |
|     | Isotonic Regression | 0.0199±0.0007    | 0.0157±0.0005    |
| 9%  | Histogram Binning   | 0.0194±0.0006    | 0.0153±0.0005    |
|     | Platt Scaling       | 0.0193±0.0008    | 0.0148±0.0007    |
|     | Scaling-Binning     | 0.0191±0.0008    | 0.0147±0.0007    |
|     | Isotonic Regression | 0.0190±0.0006    | 0.0147±0.0005    |

Table 21: Average and standard errors of MCE on the Avazu dataset

|  |  | Original | VAD+ |
|---|---|---|---|
| 3% | Histogram Binning | 0.0636±0.0022 | 0.0571±0.0022 |
|  | Platt Scaling | 0.0611±0.0017 | 0.0566±0.0018 |
|  | Scaling-Binning | 0.0611±0.0014 | 0.0554±0.0017 |
|  | Isotonic Regression | 0.0663±0.0021 | 0.0599±0.0021 |
| 5% | Histogram Binning | 0.0544±0.0014 | 0.0486±0.0014 |
|  | Platt Scaling | 0.0524±0.0019 | 0.0471±0.0019 |
|  | Scaling-Binning | 0.0515±0.0013 | 0.0463±0.0015 |
|  | Isotonic Regression | 0.0527±0.0013 | 0.0483±0.0015 |
| 7% | Histogram Binning | 0.0485±0.0015 | 0.0420±0.0013 |
|  | Platt Scaling | 0.0489±0.0016 | 0.0432±0.0016 |
|  | Scaling-Binning | 0.0457±0.0012 | 0.0409±0.0015 |
|  | Isotonic Regression | 0.0480±0.0014 | 0.0428±0.0016 |
| 9% | Histogram Binning | 0.0469±0.0014 | 0.0401±0.0014 |
|  | Platt Scaling | 0.0425±0.0014 | 0.0381±0.0015 |
|  | Scaling-Binning | 0.0429±0.0015 | 0.0374±0.0015 |
|  | Isotonic Regression | 0.0454±0.0014 | 0.0389±0.0014 |

