# OpenReview forum: "Calibration Matters: Tackling Maximization Bias in Large-scale Advertising Recommendation Systems"
_ICLR.cc/2023/Conference — ICLR 2023 poster_

### Official Review · Reviewer_imJk · 2022-10-20

**Confidence:** 3
**Correctness:** 4
**Technical Novelty And Significance:** 3
**Empirical Novelty And Significance:** 2
**Recommendation:** 8

**Clarity, Quality, Novelty And Reproducibility:**

Clarity: this paper is very well motivated and written. It is easy to follow and a good read.

Quality: this work is of high quality. The problem is well motivated and illustrated. The authors theoretically quantify the maximization bias and provide a theoretically proven algorithmic solution to tackle it.

Novelty: this paper provides theoretical insights and proposes a novel meta-algorithm to address an important problem in large-scale ads recommendation system.

Reproducibility: the authors demonstrate the effectiveness of the proposed algorithm using a state-of-the-art recommendation neural network model on a large-scale real-world dataset.


**Strength And Weaknesses:**

Strength:
The paper is well motivated and clearly written, with several highlights:
1.	They theoretically quantify the maximization bias in a supervised learning settings in recommendation systems. They prove theorems and corollaries that offer insights on the source of calibration error and the development of their novel algorithm.
2.	Their proposed algorithm is theoretically supported and efficient, which makes it practical for any machine learning methods and can be used in tandem with any existing calibration methods. They also prove that it is robust to covariate shifts between training and test sets, which is common in modern recommendation systems.
3.	The authors conducted extensive numerical experiments to demonstrate the effectiveness of the proposed meta-algorithm.

Weaknesses:
1.	As the authors mentioned, none of the baseline calibration methods they chose to benchmark against explicitly considers the maximization bias and thus fails to perform well in their setting. It would make their results more convincing if they can show some comparison results against existing calibration methods that explicitly considers the maximization bias such as double learning or cross-validation estimators. Even though these alternative estimators may not be practical in a large-scale ads recommendation system, they could still serve as valuable baselines and offer additional insights.


**Summary Of The Paper:**

In this work, the authors theoretically quantify the maximization bias in generalized linear models with Gaussian distributions. In practice, the maximization bias often manifests itself in the calibration of online advertising recommendation system. The authors propose a variance-adjusting de-biasing meta-algorithm that is able to mitigate the maximization bias problem, and is robust to covariate shifts between training and test sets that are common in modern recommendation systems. The algorithm can be used in tandem with other calibration methods without compromising the ranking performance nor increasing online serving overhead. The authors demonstrate the effectiveness of the proposed algorithm in both synthetic datasets using a logistic regression model and a large-scale real-world dataset using a state-of-the-art recommendation neural network model.

**Summary Of The Review:**

Overall, this is a high-quality paper written with clarity. Its main contribution includes theoretically quantifying the maximization bias in a supervised learning settings in recommendation systems; the development of a novel, robust, and practical algorithm that optimizes calibration; the demonstration of the effectiveness of the meta-algorithm. Their results can be strengthened by showing comparison results against existing calibration methods that explicitly considers the maximization bias such as double learning or cross-validation estimators.

---

> ### Author Response · Authors · 2022-11-10
> **Response to Reviewer imJk**
>
> We would like to thank the reviewer for the detailed feedback and valuable suggestions. We have studied the comments carefully, and please see our responses to the reviewer’s comments as follows:
>
> **Q1: As the authors mentioned, none of the baseline calibration methods they chose to benchmark against explicitly considers the maximization bias and thus fails to perform well in their setting. It would make their results more convincing if they can show some comparison results against existing calibration methods that explicitly considers the maximization bias such as double learning or cross-validation estimators. Even though these alternative estimators may not be practical in a large-scale ads recommendation system, they could still serve as valuable baselines and offer additional insights.**
>
> A1: Thanks for your suggestions! We added the double learning method following your suggestions and below are the results. The results are expected: Double Learning method gives near perfect Calibration Error and reduces ECE.
>
> | alpha | Methods         | Calibration Error | ECE         |
> | ----- | --------------- | ----------------- | ----------- |
> | 2%    | Vanilla         | 3.226±0.413       | 0.029±0.002 |
> | 2%    | Double Learning | \-0.495±0.515     | 0.024±0.002 |
> | 5%    | Vanilla         | 3.821±0.523       | 0.029±0.003 |
> | 5%    | Double Learning | \-0.238±0.623     | 0.023±0.002 |
> | 7%    | Vanilla         | 3.904±0.572       | 0.028±0.003 |
> | 7%    | Double Learning | \-0.181±0.674     | 0.023±0.002 |
> | 10%   | Vanilla         | 3.941±0.631       | 0.027±0.003 |
> | 10%   | Double Learning | \-0.108±0.742     | 0.023±0.002 |

---

### Official Review · Reviewer_1fZs · 2022-10-22

**Confidence:** 4
**Correctness:** 3
**Technical Novelty And Significance:** 3
**Empirical Novelty And Significance:** 2
**Recommendation:** 6

**Clarity, Quality, Novelty And Reproducibility:**

Overall, the paper is well written and easy to follow. The authors studied a new source of bias in Ads ranking.

**Strength And Weaknesses:**

Strength
1. The paper studied an important and practical problem in real-world Ads ranking systems.
2. The proposed method is well-justified under mild conditions from GLM.
3. The algorithm is simple to implement without additional inference cost.

Weakness
1. The results in Theorem 4.1 and Corollary 4.2 seem to suggest \alpha does not play a role in deciding the \lambda parameter in the algorithm. Consider the extreme case \alpha=100% which is the original distribution; if there is no model bias, the algorithm should be a no-op and not change the prediction at all.
2. Many real-world Ads ranking systems utilize online learning of model parameters. It would be interesting to see how the method can be generalized to online learning settings for more practical applications.
3. It would be better if the authors could test the proposed methods on several other Ads ranking datasets. It would also help to experiment on a few different model structures. Moreover, an empirical validation of the Gaussian assumption would be informative.
4. For the empirical evaluation, it would be better to have the results on \alpha=100% to measure just the model bias of vanilla and other methods.
5. For the evaluation results, why does the CE and ECE have different ordering and different performance by varying \alpha. Intuitively, as \alpha becomes bigger, the maximization bias becomes smaller and the comparing algorithm should have better performance. Also, though the proposed method does change accuracy, it would be better to provide the metrics like log loss and AUC as a reference for model performance.


**Summary Of The Paper:**

In this paper, the authors study the problem of correcting maximization bias in advertising recommendation systems. The authors identify maximizing bias from the operation of ranking and select top a% ads and it presents even the model is well-calibrated on the entire distribution. The authors proposed a simple correction algorithm motivated via rigorous analysis of GLM with mild assumptions. Experiments on both synthetic and real-world datasets are carried out with comparison to other calibration methods.

**Summary Of The Review:**

The paper studied a new source of bias in Ads ranking and proposed a simple and just-justified solution. However, there is some concern in empirical evaluation and practicality in real-world application.

---

> ### Author Response · Authors · 2022-11-10
> **Response to Reviewer 1fZs part 1**
>
> We would like to thank the reviewer for the detailed feedback and valuable suggestions. We have studied the comments carefully, and please see our responses to the reviewer’s comments as follows:
>
> **Q1: The results in Theorem 4.1 and Corollary 4.2 seem to suggest \alpha does not play a role in deciding the \lambda parameter in the algorithm. Consider the extreme case \alpha=100% which is the original distribution; if there is no model bias, the algorithm should be a no-op and not change the prediction at all.**
>
> A1: Thanks for the great observations. Yes, \alpha does not play a role in deciding the \lambda parameter, which can be considered as an advantage. For the extreme case with alpha = 100% you mentioned, we agree the *calibration* would not change. However, individual predictions should be corrected towards the mean due to the maximization bias. Since the calibration procedure lowers high values and increases the low values, the total calibration would remain exactly the same if $\phi$ is linear (the total calibration would remain roughly the same if $\phi$ is non-linear because nonlinearity would only create second-order differences). Regarding the case with \alpha=100% and no model bias, the algorithm should still change the predictions. Consider an extreme case where all ads have the identical true ctr 0.5, and our ML model is equally likely to assign an estimated value of 0.6 or 0.4 to every ad independently, so there is no model bias. Applying our method will push the predictions towards mean, which is the true ctr 0.5, thus producing better predictions than the original unbiased model predictions.
>
> **Q2: Many real-world Ads ranking systems utilize online learning of model parameters. It would be interesting to see how the method can be generalized to online learning settings for more practical applications.**
>
> A2: Thanks for the suggestions. We agree that applying our method to an online learning setting is one of the interesting follow-ups. We hypothesize that the lambda value won’t change too much in the real world online learning setting, so we only need to update the lambda value from time to time. We will verify and investigate it systematically in our future research.
>
> **Q3: It would be better if the authors could test the proposed methods on several other Ads ranking datasets. It would also help to experiment on a few different model structures. Moreover, an empirical validation of the Gaussian assumption would be informative.**
>
> A3: Figure 5 (in Appendix B.2) justified Gaussian assumption. From the figure, the last-layer neurons (logits) in the neural networks look Gaussian. By the Kolmogorov-Smirnov test, we fail to reject the null hypothesis that the empirical distribution is Gaussian at the 5% significance level, which is consistent with the shape of distribution.
>
> In the literature, we found the Avazu dataset (https://www.kaggle.com/c/avazu-ctr-prediction) as another commonly used Ads Ranking dataset. We are working on testing the proposed methods on the Avazu dataset with different model structures. We will update as soon as we get systematic results.

---

> ### Author Response · Authors · 2022-11-10
> **Response to Reviewer 1fZs part 2**
>
> **Q4: For the empirical evaluation, it would be better to have the results on \alpha=100% to measure just the model bias of vanilla and other methods.**
>
> A4: Thanks for the suggestion. To begin with, we need to formally define the model bias. Let $f$ be a random predictor learned for the training data and let $f^*$ be the underlying true predictor. Then, we define the model bias for an ad with feature $x$ as $ \mathbb{E}[f(x)]-f^*(x)$, where  $x$ is fixed and the expectation is taken over $f$.
>
> Therefore, the extreme case with $ \alpha=100\\%$ corresponds to the sum of the model biases of all ads. However, we think it is better to thoroughly examine the model bias, especially for those ads with high $f^*(x)$. Since we do not know the underlying true $f^*(x)$ for the Criteo dataset, we develop the following procedure to approximate $f^*(x)$ and compute model biases in various regions.
>
> To estimate the model bias for high true ctr ads, we use the following way and your feedback will be appreciated. We train 80 models with different random initialization seed and data orders. We then use the average prediction of 80 models as our predictions with low variance. The model bias can be approximated by the calibration error of this average prediction. And below we report the calibration errors for the vanilla method. Here we choose top alpha predictions based on the average prediction (note this still suffers from maximization bias in theory, but in practice the maximization bias is very small since the average prediction has very low variance). As we can see, the model bias is relatively small, and even has a small downward bias.
>
> | alpha   | 2%      | 3%      | 4%      | 5%      | 6%      | 7%      | 8%      | 9%      | 10%     |
> | ------- | ------- | ------- | ------- | ------- | ------- | ------- | ------- | ------- | ------- |
> | Vanilla | \-0.70% | \-0.31% | \-0.34% | \-0.13% | \-0.10% | \-0.14% | \-0.22% | \-0.20% | \-0.16% |
>
> **Q5: For the evaluation results, why does the CE and ECE have different ordering and different performance by varying \alpha. Intuitively, as \alpha becomes bigger, the maximization bias becomes smaller and the comparing algorithm should have better performance. Also, though the proposed method does change accuracy, it would be better to provide the metrics like log loss and AUC as a reference for model performance.**
>
> A5: We provided log loss results in Table 11 in the Appendix B.2. And indeed, we see a log loss improvement. AUC will remain unchanged after applying our method, since our method won’t change the ordering of predictions.
>
> CE and ECE have different ordering because the calculation is different. To calculate CE, we take the average of each model’s calibration errors, which can be positive or negative. To calculate ECE, we take the average of each model’s ECE, which has to be positive.
>
> As \alpha becomes bigger, the maximization bias becomes smaller as you pointed out. However, the calibration error should remain roughly the same because calibration error can be viewed as the ratio between the average calibration error and the average true probability. As \alpha increases, the average true probability is also decreasing. This is consistent with Figure 2.a. And ECE is roughly proportional to the maximization bias. Therefore, we see a downward trend for the comparing algorithms in Figure 2.b.

---

> ### Author Response · Authors · 2022-11-14
> **Response to Reviewer 1fZs part 3**
>
> Q3: It would be better if the authors could test the proposed methods on several other Ads ranking datasets. It would also help to experiment on a few different model structures.
>
> A3: We tested our method on Avazu dataset (https://www.kaggle.com/c/avazu-ctr-prediction) using a different model structure xDeepFM open sourced in https://deepctr-doc.readthedocs.io/en/latest/deepctr.models.xdeepfm.html. We took the first 10M rows from the dataset to speed up the model training, and we trained 60 models (we replicate the experiments 30 times). We have updated the results in Appendix and the model training code for these additional experiments in the “Supplementary Material”. Our method still works well in this case. In particular, all the calibration methods in tandem with VAD achieve better performance than the sole calibration methods alone, which is consistent with results in our paper.
>
> **Calibration Error Results**:
>
> | alpha | Methods             | Original      | VAD+            |
> | ----- | ------------------- | ------------- | --------------- |
> | 2%    | Histogram Binning   | 2.2627±0.1405 | \-0.0419±0.1443 |
> | 2%    | Platt Scaling       | 2.2844±0.221  | \-0.0338±0.2111 |
> | 2%    | Scaling-Binning     | 2.3425±0.2255 | 0.0134±0.2152   |
> | 2%    | Isotonic Regression | 2.362±0.1436  | 0.0436±0.1476   |
> | 5%    | Histogram Binning   | 5.1018±0.2644 | 2.764±0.2659    |
> | 5%    | Platt Scaling       | 5.1861±0.2685 | 2.8203±0.267    |
> | 5%    | Scaling-Binning     | 5.2733±0.2721 | 2.9061±0.2707   |
> | 5%    | Isotonic Regression | 5.1651±0.266  | 2.7975±0.2662   |
> | 7%    | Histogram Binning   | 5.7854±0.2754 | 3.4662±0.2803   |
> | 7%    | Platt Scaling       | 5.734±0.2792  | 3.3876±0.2822   |
> | 7%    | Scaling-Binning     | 5.7618±0.2794 | 3.4175±0.2828   |
> | 7%    | Isotonic Regression | 5.6671±0.2714 | 3.3186±0.2777   |
> | 10%   | Histogram Binning   | 6.1142±0.2532 | 3.8368±0.2579   |
> | 10%   | Platt Scaling       | 5.9183±0.2855 | 3.6146±0.2875   |
> | 10%   | Scaling-Binning     | 6.0124±0.2887 | 3.7117±0.291    |
> | 10%   | Isotonic Regression | 5.947±0.2537  | 3.637±0.2562    |
>
> **ECE results**:
>
> | alpha | Methods             | Original      | VAD+          |
> | ----- | ------------------- | ------------- | ------------- |
> | 2%    | Histogram Binning   | 0.022±0.0005  | 0.0211±0.0005 |
> | 2%    | Platt Scaling       | 0.0223±0.0005 | 0.0211±0.0006 |
> | 2%    | Scaling-Binning     | 0.0216±0.0004 | 0.0205±0.0005 |
> | 2%    | Isotonic Regression | 0.0224±0.0005 | 0.0211±0.0004 |
> | 5%    | Histogram Binning   | 0.0208±0.0007 | 0.0173±0.0005 |
> | 5%    | Platt Scaling       | 0.0211±0.0008 | 0.0172±0.0007 |
> | 5%    | Scaling-Binning     | 0.0213±0.0008 | 0.0173±0.0007 |
> | 5%    | Isotonic Regression | 0.021±0.0007  | 0.0171±0.0005 |
> | 7%    | Histogram Binning   | 0.0202±0.0007 | 0.0163±0.0006 |
> | 7%    | Platt Scaling       | 0.0201±0.0008 | 0.0157±0.0007 |
> | 7%    | Scaling-Binning     | 0.02±0.0008   | 0.0157±0.0007 |
> | 7%    | Isotonic Regression | 0.0199±0.0007 | 0.0157±0.0005 |
> | 10%   | Histogram Binning   | 0.0188±0.0006 | 0.0147±0.0005 |
> | 10%   | Platt Scaling       | 0.0184±0.0008 | 0.0142±0.0006 |
> | 10%   | Scaling-Binning     | 0.0185±0.0008 | 0.0141±0.0007 |
> | 10%   | Isotonic Regression | 0.0185±0.0006 | 0.0142±0.0005 |

---

### Official Review · Reviewer_hQai · 2022-10-25

**Confidence:** 4
**Correctness:** 4
**Technical Novelty And Significance:** 3
**Empirical Novelty And Significance:** 3
**Recommendation:** 6

**Clarity, Quality, Novelty And Reproducibility:**

Clarity:
The paper is fairly well-written and explained.

Quality: good quality.

Novelty: I believe this method is novel, including the theoretical analysis. It is a compelling direction to consider how to make use of unlabeled data in a practical problem.

Reproducibility: The code is made available by the authors (I did not run it) and the datasets are publically available so it should be reproducible.


**Strength And Weaknesses:**

Strengths:
1. This is an important, practical problem.

2. The paper is fairly well-explained although there are areas for improvement (particularly the jump from the theory with alpha and the practical algorithm).

3. The offline results are compelling including on real datasets (both decreasing bias and improve accuracy).

Weaknesses:
1. This may be of more interest in a venue focused on online advertising.

2. It would be more compelling to try an online experiment (but it may not be possible for the authors).

3. The assumption that you can practically obtain all candidates may not be realistic in many industry settings  where that set is very large and constantly changing.

**Summary Of The Paper:**

Calibration is important for models used for predictions in online advertising systems. It can be challenging because there is a form of adverse selection, where any over-predictions are more likely to result in these ads being shown. There is also a challenge because the training data is typically the ads that were shown and so is not a random selected subset of ads.

This work proposes a method for a calibration step making use of an unlabelled test set that is unbiased (containing all possible candidates). Then it creates multiple estimators (e.g. through bootstrapping) and estimates the variance of their predictions on the test set. This is used to calibrate the predictions. There is both theoretical analysis and (offline) experiments of the approach.


**Summary Of The Review:**

A good paper on an interesting topic. It may be too specialized for the ICLR audience to appreciate. I have some concerns about the practicality of the approach and it would be more compelling to see this tried online.

---

> ### Author Response · Authors · 2022-11-10
> **Response to Reviewer hQai**
>
> We would like to thank the reviewer for the detailed feedback and valuable suggestions. We have studied the comments carefully, and please see our responses to the reviewer’s comments as follows:
>
>
> **Q1: This may be of more interest in a venue focused on online advertising.**
>
> A1: Thanks for the suggestion. Let us respectfully add that, although we were inspired by online advertising, the problem is rather general and we can handle maximization bias as long as there is a selection procedure. For example, we think similar ideas could be potentially used for correcting the maximization bias in Q-learning as an alternative method of double learning. In the ”Related Work” section, we pointed out the importance of maximization bias in RL literature.
>
> **Q2: It would be more compelling to try an online experiment (but it may not be possible for the authors).**
>
> A2: We would like to thank you for the great suggestions and for being understanding. We agree that an online experiment would be more convincing. We would like to collaborate with industries to try this idea in an online experiment in the future.
>
> **Q3: The assumption that you can practically obtain all candidates may not be realistic in many industry settings where that set is very large and constantly changing.**
>
> A3: Thanks for the question! We do not need all candidates, but only need samples from the unlabeled candidate set of reasonable size. And as we only need to estimate mean and variances, we can sample recent data points in the large (and potentially constantly changing) candidate set to get sufficiently accurate estimates. Therefore, large and constantly changing sets are usually not a practical concern.

---

> > ### Comment · Reviewer_hQai · 2022-11-14
> > **Response**
> >
> > I appreciate the authors responding to my review, particularly Q3. I would suggest since that's an important practical point it might be worth highlighting in the paper.

---

> > > ### Author Response · Authors · 2022-11-15
> > > **Thanks for the suggestion**
> > >
> > > Thanks for your suggestion! We have updated our manuscript. Please check the paragraph below Algorithm1 on Page 6.

---

### Official Review · Reviewer_MXBz · 2022-10-27

**Confidence:** 4
**Correctness:** 3
**Technical Novelty And Significance:** 3
**Empirical Novelty And Significance:** 3
**Recommendation:** 6

**Clarity, Quality, Novelty And Reproducibility:**

The paper presentation can be improved in my opinion:
- The "Example 1" outlining a case of maximization bias isn't very clear. Should be rephrased/clarified or just replaced with a better example.
- Figure 1 flowchart is unclear and unhelpful.
- MCE: isn't used in main body, please move to Appendix.
- The claim that "h'(t) has the same order as ..." should be clarified and formalized in a proof in the Appendix.
- Algorithm 1: notation with f_i^\ell and f_i is quite confusing. I would suggest to define the predictors to be \psi(f_i(x)) and clarify that f_i can be obtained either by inverting \psi or looking at the last layer of the NN.
- Can you give some intuitive explanation on \sigma_f / \sigma_Y ? When is this ratio close to 1 or close to 0? Also why can't it be more than 1?

**Strength And Weaknesses:**

Pros:
- Analysis for the GLM case is sound. The analysis relies on the fact that the ad selection is so that only the top-scorer ads really matter.
- Algorithm is simple and can be applied to any ML model.

Cons:
- Requires multiple training runs (even though the authors suggest a single extra run might suffice)
- The experiments rely on adding covariate shift (both for the synthetic and Criteo dataset), so it is unclear what the benefits of using this approach are in the case where there's little-to-none covariate shift.
- The debiasing algorithm relies on learning the X margin in the test set.


**Summary Of The Paper:**

The authors focus on the problem of calibration in the online advertising setting, where the ML model has to predict the probability that an ad will be clicked. These probabilities are then used in the ad exchange to inform the ranking and selection of ads, but also to estimate the value of the chosen ad to charge the advertisers.

Specifically, the paper focuses on tackling maximization bias when calibrating the model. This maximization bias occurs in this setting because the same dataset is used to determine the maximizing action and to estimate its value.

The first contribution of the authors is a theoretical analysis of the problem in the case of generalized linear models with Gaussian features. This analysis leads to a debiasing term to adjust for this bias. This debiasing term is learned by retraining the model on bootstrap samples (or using different random seeds.)

The debiasing logic is then extrapolated to a more general algorithm that can be applied to any ML model.

The experimental section encompasses a synthetic study of a logistic regression model with Gaussian features (to emulate the analysis of the GLM case), as well as a deep learning recommendation model (DLRM) on the Criteo Ad dataset.

**Summary Of The Review:**

I do see some novelty and appreciate the formal analysis (even though it only applies to GLM with Gaussian features).

My recommendation is a weak accept.

---

> ### Author Response · Authors · 2022-11-10
> **Response to Reviewer MXBz**
>
> We would like to thank the reviewer for the detailed feedback and valuable suggestions. We have studied the comments carefully, and please see our responses to the reviewer’s comments as follows:
>
> **Q1: Regarding the benefits of using our approach in the case where there's little-to-none covariate shift**
>
> A1: Thanks for the great question! In the real-world ads recommendation system setting, there is usually a covariate shift between training data and test data, as pointed out in the introduction. So this paper primarily focuses on the covariate shift setting.
>
> Our method also reduces maximization bias when there is no covariate shift, as the theory suggests, which was validated by experiments on synthetic dataset using Logistic Regression, shown below. We didn’t include this result in the paper due to page limits.
>
> | alpha | Methods | Calibration Error | ECE         |
> | ----- | ------- | ----------------- | ----------- |
> | 2%    | Vanilla | 8.911%±0.294%     | 0.133±0.001 |
> | 2%    | VAD     | \-0.275%±0.305%   | 0.13±0.001  |
> | 5%    | Vanilla | 7.673%±0.22%      | 0.087±0.0   |
> | 5%    | VAD     | \-0.595%±0.24%    | 0.083±0.0   |
> | 7%    | Vanilla | 7.214%±0.201%     | 0.074±0.0   |
> | 7%    | VAD     | \-0.566%±0.222%   | 0.07±0.0    |
> | 10%   | Vanilla | 6.676%±0.182%     | 0.063±0.0   |
> | 10%   | VAD     | \-0.674%±0.2%     | 0.059±0.0   |
>
> In our experiments using Criteo dataset under no covariate shift setting, our method is still able to correct maximization bias on top of vanilla predictions. In this case, the model bias is the dominating factor (instead of maximization bias), so correcting maximization bias won’t have a significant impact. We found out that some traditional calibration methods are able to do well under no covariate shift setting, and applying our method on top of other calibration methods won’t change the predictions under no covariate shift setting, as pointed out in the Section 6.2, “Baseline Calibration Methods” section.
>
> So in any case, applying our method on top of other calibration methods will make the results better or at least on par.
>
>
> **Q2: The debiasing algorithm relies on learning the X margin in the test set.**
>
> A2: Indeed, the algorithm relies on some property of the marginal distribution of X. More specifically, as shown in Algorithm 1, we make use of the mean, variances and conditional variances of test data’s predictions, which can be easily calculated from the test data. It’s worth noting that our method doesn’t require estimation of the whole marginal distribution of X. Furthermore, computing those statistics only requires unlabeled test data, which is easy to obtain in practice.
>
> **Q3: Can you give some intuitive explanation on \sigma_f / \sigma_Y ? When is this ratio close to 1 or close to 0? Also why can't it be more than 1?**
>
> A3: Thanks for your question. \sigma_f / \sigma_Y is the proportion of the predictor randomness to the total test set estimation randomness, and it cannot be larger than 1, which we formally justify below:
>
> Let $f^{\ast }(x)$ be the underlying true predictor. Then, we can generalize
> the decomposition (Equation (7)) as
> $$
> \mathrm{Var}(f(X))=\mathrm{Var}\left( f^{\ast }(X)\right) +\mathbb{E}\left[
> \mathrm{Var}\left( f(X)|X\right) \right] ,
> $$
> provided that the learned predictor $f$ $\ $is unbiased, i.e., $\mathbb{E}%
> \left[ f(x)\right] =f^{\ast }(x)$ for all $x.$ Then, in our notation,
> $$
> \hat{\sigma}_{\hat{Y}}^{\ell }=\mathrm{Var}(f(X)),\text{ and }\hat{\sigma}%
> _{f}^{\ell }=\mathbb{E}\left[ \mathrm{Var}\left( f(X)|X\right) \right] .
> $$
>
> Therefore,  $\\hat{\\sigma}_{f}^{\\ell} \leq \\hat{\\sigma}^{\\ell}\_{\\hat{Y}}$ as $
> \mathrm{Var}\left( f^{\ast }(X)\right) \geq 0$ and thus the ratio cannot be
> more than 1.
>
> When there is little randomness in the predictor (i.e. the predictor’s variance is low), the maximization bias is small and thus lambda is close to 1. On the other hand, if there is little randomness with the underlying test distribution, the test set estimation variance is roughly equal to the predictor variance. This case corresponds to the scenario with a low signal to noise ratio. And we have the ratio close to 0, making the calibrated probability close to the test mean.  It is desired as the true values in the test set centers on the mean.
>
> **Q4: Regarding paper presentation**
>
> A4: Thanks for your comments and suggestions. We have updated our paper in light of your comments and suggestions. In particular, we rephrase Example 1, update the Figure 1 flowchart, move MCE to the appendix, formally justify the claim that "h'(t) has the same order as ..." in Lemma A.2, and change the notation in Algorithm 1.

---

### Decision · Program_Chairs · 2023-01-20

**Decision:**

Accept: poster

**Justification For Why Not Higher Score:**

This paper is not really about learning representations.

**Justification For Why Not Lower Score:**

All reviewers and the AC agree that the problem tackled is important, and the reviewers believe that the main theorem is correct.

**Metareview: Summary, Strengths And Weaknesses:**

All reviewers are positive about this paper, and the authors have addressed suggestions for improvement well.

As the AC, I have a few concerns or suggestions that I hope the authors will address in the final version.

1. Maximization bias is essentially the same thing as the so-called winner's curse, which arises in auctions in general. To make the paper more exciting and useful to a broader audience, discuss this connection.

2. The authors recommend S=2 at the top of page 7. This makes the VAD algorithm reminiscent of the double learning (DL) that is well known in RL. Please discuss.

3. The paper says that DL is "not applicable to large-scale ads recommendation systems since it will double online serving costs, and online serving efficiency is essential to recommendation system performances." But the second model in DL is needed just once, to estimate the value of the action chosen using the first model. In contrast, the first model must be invoked many times, on each candidate action, so DL does not double online serving costs. DL seems simple and affordable in practice, and in industry simplicity is of high value. One of the authors' responses shows that DL works very well in practice.

4. I am not convinced that the authors have really absorbed correctly and cited correctly the references. Consider the 2012 paper "Predicting accurate probabilities with a ranking loss." The current submission cites this paper with context "accurately estimate the value of the chosen ads" and "isotonic regression". Both these topics are mentioned in the 2012 paper, but it is definitely not the first or best reference for either topic, and in fact *choosing* ads is not mentioned at all.

**Note From Pc:**

if the above contains the word "oral" or "spotlight" please see: "oral" presentation means -> notable-top-5% and "spotlight" means -> notable-top-25%. As stated in our emails, we are disassociating presentation type from AC recommendations

**Summary Of Ac-Reviewer Meeting:**

No meeting.

---

> ### Author Response · Authors · 2023-02-28
> **Response to the AC**
>
> We would like to thank the AC for the detailed feedback and valuable suggestions.
>
> Q1: Maximization bias and winner's curse
>
> A1: We’ve included a new paragraph discussing the connection in the introduction section. Thanks for the great suggestion.
>
> Q2: Regarding VAD algorithm and Double Learning (DL)
>
> A2: We’ve discussed “Double Learning” method in the second paragraph of “Related Work” section. And we fully agree that “industry simplicity is of high value”. Based on our industry experiences, shipping methods like DL in industry is not trivial. If we use the first model to rank and the second model to estimate the values, we have to do it sequentially, increasing the system latency. And in the ads recommendation system setting, we have a fixed time budget to respond to a user request. Increasing system latency means we need to run models on less ads, which will usually result in less revenue.
>
> In addition, serving cost does not only refer to “computation cost”. Industrial Ads Recommendation Models usually need large memory (because of the huge embedding tables). Double Learning will double the serving memory cost, which is usually not affordable.
>
> Q3: Citations
>
> A3: Thanks for pointing them out. We’ve corrected the citation issues in our final version.